# FLASC: Federated LoRA with Sparse Communication

## Abstract

Low-rank adaptation (LoRA) is a promising method for finetuning models in communication-constrained settings such as cross-device federated learning (FL). Prior work has explored ways to improve the efficiency of LoRA in federated settings by imposing additional sparsity constraints. However, as we show, existing methods for sparse LoRA not only harm accuracy but can in fact *increase* overall communication costs. We instead propose **FLASC**, a simple approach with two key components: First, **FLASC** combines LoRA with *sparse communication*, which outperforms baselines such as using a lower LoRA rank or pruning LoRA weights. Second, **FLASC-Search** efficiently searches the space of sparsity-and-rank configurations by iteratively comparing pairs of configurations and increasing either the rank or density. Across four FL datasets, we demonstrate that **FLASC** outperforms existing sparse LoRA methods with up to 20% higher accuracy or 10× less communication. Our work highlights the importance of considering the constraints of existing efficient finetuning methods and provides a simple and competitive baseline for future work in federated finetuning.

## 1 Introduction

As pretrained models continue to advance state-of-the-art performance in a variety of domains, it is critical to develop methods to efficiently finetune models in low-resource settings. In this work, we consider the cross-device federated learning (FL) setting which seeks to train models across a network of decentralized and heterogeneous clients (McMahan et al., 2017a). A major bottleneck in FL is the cost of *uploading model updates* to the server, which can significantly slow down finetuning (Konečný et al., 2017).

Recently, parameter-efficient finetuning (PEFT) has emerged as an effective way to reduce costs in both centralized and federated settings (Houlsby et al., 2019; Hu et al., 2021; Zhang et al., 2023c). We focus on *low-rank adaptation* (LoRA), a popular method that injects trainable low-rank adapters into a model and freezes the pretrained backbone. To further improve LoRA, prior works in centralized ML apply sparsity by pruning individual entries or groups of weights during training (Wu & Chen, 2022; He et al., 2022; Zhang

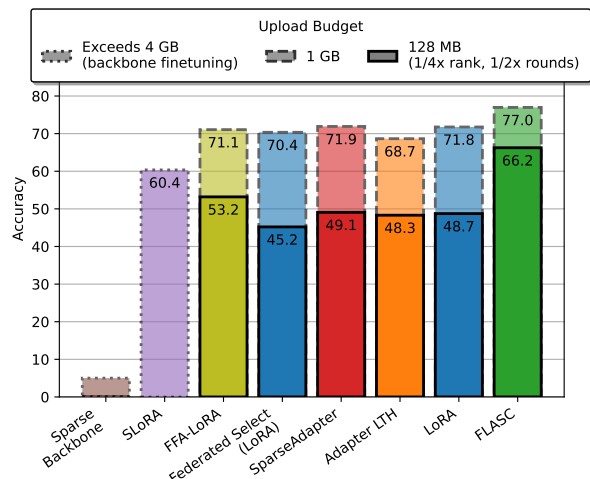

**Figure 1:** We train FL models on 20NewsGroups while applying sparsity and/or LoRA. LoRA itself can be more efficient than existing methods, while **FLASC** outperforms LoRA in constrained settings.

et al., 2022; 2023a). Similar works in the FL setting also consider sparsity, but focus on improving LoRA in heterogeneous and private FL settings (Cho et al., 2023; Babakniya et al., 2023a; Sun et al., 2024). Unfortunately, prior works fail to address the key challenge of *configuring LoRA*, as they must still tune a rank hyperparameter and additional hyperparameters related to sparsity—costs that can quickly overshadow the savings attained from sparsity. Further, even with optimal hyperparameters, these methods can perform worse than simply using LoRA with an equal communication budget (see Fig. 1).

To address these issues, we present a seemingly overlooked baseline in communication-efficient federated finetuning: **FLASC** (**F**ederated **L**ow-Rank **A**daptation with **S**parse **C**ommunication). Our method has two key contributions: First, **FLASC** computes dense local updates while only communicating sparse updates, which we show greatly improves performance over approaches that reduce the LoRA rank or freeze LoRA weights. Second, we present a hyperparameter search method **FLASC-S** (**FLASC**-**S**earch) that makes it practical to realize the benefits of sparsity. This method relies on a key tradeoff observed when applying sparsity via **FLASC**: reducing the rank is more helpful for reducing redundant communication under a large budget, while sparsity is more helpful for improving utility under a small budget. In terms of both a single training run and the total cost of tuning, our method significantly improves the communication-efficiency of federated LoRA. Furthermore, we find that **FLASC** is equally robust to LoRA in terms of other common FL concerns such as heterogeneity and privacy. Overall, our work makes the following contributions:

1. We propose **FLASC**, a simple method which reduces the communication cost of LoRA by applying unstructured sparsity to upload communication. **FLASC** can reduce communication of LoRA by up to 10× without harming accuracy, or improve accuracy by up to 20% under the same communication budget. Despite its simplicity, **FLASC** has been overlooked as a baseline by prior work in efficient FL.

2. We study the interaction between LoRA rank and sparse communication in **FLASC**. We find that sparsity is more important at small communication budgets, while reducing the rank is more important at large budgets. Based on this observation, we propose **FLASC-S**, an efficient tuning method which sequentially tunes the density followed by rank, allowing **FLASC** to save communication even when considering the cost of hyperparameter tuning.

3. We conduct extensive experiments which show that both LoRA and **FLASC** are effective when deployed in settings where other common challenges in FL are present, such as data/systems heterogeneity or differential privacy. In fact, we find that **FLASC** outperforms other sparsity and freezing-based methods that were designed specifically for these concerns.

## 2 Related Work

**Communication-efficient federated learning.** Upload communication is a key bottleneck in cross-device FL (Konečný et al., 2017). While large pretrained models can significantly improve utility in FL applications (Radford et al., 2018; Nguyen et al., 2022), these models present new challenges with communication and finetuning at the edge. Many types of methods have been explored to reduce FL communication costs, including quantization (Reisizadeh et al., 2020; Ozkara et al., 2021), sparsity (Horvath et al., 2021; Stripelis et al., 2022; Isik et al., 2022; Huang et al., 2022), parameter-efficient finetuning (Chen et al., 2023), and their combinations (Caldas et al., 2018b; Ro et al., 2022; Babakniya et al., 2023a).

**Sparsity and pruning.** In centralized ML, *pruning* is a popular area of research that aims to reduce compute and storage costs by sparsifying and freezing weights (Frankle & Carbin, 2018; Dettmers & Zettlemoyer, 2019; Sung et al., 2021). Early studies of sparsity in distributed and federated learning focus on the use of *sparse communication* only (Basu et al., 2019; Gao et al., 2021; Mitra et al., 2021). More recent FL methods utilize pruning by sending a sparse model to clients and finetuning in a sparse-to-sparse manner (Shah & Lau, 2021; Qiu et al., 2021; Bibikar et al., 2022; Babakniya et al., 2023b). However, a limitation is that sparsity methods are mainly studied in the context of full model finetuning, even though PEFT methods are already very communication-efficient. Therefore, we study how to apply *unstructured* (weight-level) sparsity to LoRA. We chose unstructured rather than structured (block-level) sparsity due to the limited utility of structured sparsity in both centralized (Liu et al., 2018) and FL settings (Caldas et al., 2018b; Cheng et al., 2022).

**Parameter-efficient finetuning (PEFT).** PEFT reduces the cost of finetuning by training a small number of parameters and freezing the rest of the model (Ding et al., 2022). In this work, we focus on *low-rank adaptation* (LoRA), a popular reparameterization-based method which has two advantages: First, LoRA parameters can be merged with the backbone after training in order to maintain the same inference costs (Houlsby et al., 2019; Hu et al., 2021). Second, LoRA tends to outperform PEFT methods which finetune the backbone (Guo et al., 2021; Zaken et al., 2022; Sung et al., 2021; Gong et al., 2022).

**Efficient LoRA.** Our work is most similar to recent works in the centralized setting that *train LoRA with unstructured sparsity* (Wu & Chen, 2022; He et al., 2022). Other works improve LoRA in various ways with sparsity (Ding et al., 2023; Liu et al., 2024; Zhang et al., 2025), quantization (Xu et al., 2023; Dettmers et al., 2024; Zhou et al., 2025), and flexibly adjusting the rank (Zhang et al., 2022; 2023a). LoRA itself can also be used to efficiently update or guide pruning of the backbone parameters (Zhao et al., 2023; Zhang et al., 2023b; Zhao et al., 2024; Ribeiro et al., 2024). In contrast to these prior works, we focus on how to make LoRA more communication-efficient in the context of FL.

**Federated LoRA.** Many works have observed that LoRA can reduce the communication cost of FL finetuning (Sun et al., 2022; Malaviya et al., 2023; Zhang et al., 2023d; Nguyen et al., 2024; Singhal et al., 2025). However, as we show, using LoRA alone is a rigid approach for communication reduction, potentially leaving a substantial amount of cost savings on the table. Our work takes an approach similar in spirit to COMPEFT, which merges compressed LoRA adapters. However, they target the application of merging adapters from multiple sources in a one-shot fashion and use more complex merging procedures (Yadav et al., 2023). In contrast, we study how to reduce both upload and download communication over multiple rounds of FL training and analyze these modifications in the context of standard FL optimization methods such as FedAvg and FedAdam (McMahan et al., 2017a; Reddi et al., 2020). Beyond communication, prior works have studied the use of LoRA with respect to FL-specific concerns such as data heterogeneity (Kim et al., 2023b; Babakniya et al., 2023a; Lu et al., 2024; Jiang et al., 2024), systems heterogeneity (Cho et al., 2023; Bai et al., 2024; Chen et al., 2025), and differential privacy (Sun et al., 2024; Bossy et al.). We show that our use of sparse uploads can reduce LoRA communication costs without harming robustness to heterogeneity and privacy, resulting in performance that matches or even exceeds the performance of these specialized methods.

Overall, **FLASC** fills in two important missing pieces in the current literature: First, while sparse communication has been widely discussed in FL, its application to LoRA has not been well-studied. Second, while many extensions to LoRA have been proposed in centralized and FL settings, we show that simple TopK sparsity is more communication-efficient than these more complex approaches.

## 3   FLASC: Federated Low-Rank Adaptation with Sparse Communication

We first introduce several sparsity baselines and then discuss benefits of **FLASC** relative to these methods. For each matrix of trainable parameters $P$, these methods maintain an unstructured (weight-level) mask $M$. Pruning-based approaches apply $M$ to all FL operations, namely communication, inference ($P \odot M$), and gradient computation ($\nabla_P \odot M$). At $d\%$ density, communicating $P$ costs $d\%$ of the original cost plus a small cost from communicating $M$. *See Appendix A.1 for details on FL optimization using FEDADAM and sparsity.*

**Sparse Backbone Baselines.** Prior work in FL focuses on training and pruning the base model. We evaluate several such methods: FEDSPARSIFY-GLOBAL (Stripelis et al., 2022) applies iterative magnitude pruning (IMP) (Renda et al., 2019). SPDST (Babakniya et al., 2023b) applies pruning-at-initialization (PaI) (Lee et al., 2018). FEDERATED SELECT maintains a dense global model and applies temporary magnitude pruning before the model is downloaded. FEDSPA (Huang et al., 2022) is similar, but applies personalized local masks.

**Federated LoRA.** LoRA is a PEFT method that freezes the pretrained backbone $W \in \mathbb{R}^{d \times k}$ and constrains its update $\Delta W \in \mathbb{R}^{d \times k}$ to be a product $BA$ where $B \in \mathbb{R}^{d \times r}$ and $A \in \mathbb{R}^{r \times k}$ are newly inserted low-rank parameters (Hu et al., 2021). To apply LoRA to FL, we set $(A, B)$ as the trainable weights for FEDADAM.

**Sparse LoRA Baselines.** While the communication cost of LoRA can be reduced by initializing $(A, B)$ with a smaller rank $r$, **"sparse LoRA"** methods obtain more expressive updates by applying the pruning-based ideas from **sparse backbone** methods to LoRA. To the best of our knowledge, there are two such baselines: ADAPTER LTH (Wu & Chen, 2022) applies IMP, while SPARSEADAPTER (He et al., 2022) applies PaI.

**FL LoRA Baselines.** Finally, we evaluate FL-specific methods designed for concerns of data heterogeneity, systems heterogeneity, and privacy. These are SLORA (Section 4.3), HETLORA (Section 4.4), and FFA-LORA (Section 4.5) respectively. However, a key limitation of these methods is that they are evaluated in terms of communication rounds. In contrast, **FLASC** is designed with communication cost as the primary concern and uses significantly less total communication. Details on these methods are in Appendix A.3.

| Method | Description | Upload ($\downarrow$) | Acc ($\uparrow$) |
|---|---|---|---|
| *Sparse backbone: Train $W$ and reduce communication by 25 or 50%* | | | |
| Full Finetune | - | 100GB | 78.0 |
| FedSparsify (Stripelis et al., 2022) | Prune a fraction of remaining parameters after model aggregation | 75GB 50GB | 65.3 5.0 |
| SPDST (Babakniya et al., 2023b) | Prune random parameters before FL; amount based on layer-wise sensitivity | 75GB 50GB | 75.6 5.0 |
| Federated Select (Charles et al., 2022) | Learn dense global model; apply a global mask before download | 75GB 50GB | 76.8 5.0 |
| FedSpa (Huang et al., 2022) | Learn dense global model and a personalized mask for each client | 75GB 50GB | 5.0 5.0 |
| *Sparse LoRA: Train $A, B$ and reduce communication by 75%* | | | |
| LoRA ($r = 16$) (Hu et al., 2021) | - | 4GB | 78.2 |
| Adapter LTH (Wu & Chen, 2022) | Prune a fraction of remaining LoRA parameters after model aggregation | 1GB | 68.7 |
| SparseAdapter (He et al., 2022) | Prune parameters before FL; based on individual parameter sensitivity | 1GB | 71.9 |
| **FLASC** (ours) | Sparsify only communication | **1GB** | **78.1** |

**Table 1:** We finetune GPT2 on 20NewsGroups for 200 rounds and reduce communication with sparsity, LoRA, or both. LoRA is much more efficient than "Sparse backbone" methods. "Sparse LoRA" methods (ADAPTER LTH, SPARSEADAPTER) can further reduce the communication of LoRA, but harm accuracy. **FLASC** only sparsifies communication and is able to match the accuracy of LoRA with 4× less communication.

**LoRA is a strong baseline.** Table 1 shows that **sparse backbone** methods are poor baselines, as they harm accuracy at mild communication reductions (75GB) or are unusable at larger reductions (50GB). In contrast, LoRA significantly reduces communication (4GB) without any drop in accuracy. Although we expect LoRA to outperform these baselines, it is surprising that LoRA performs better with almost 20× less communication. Therefore, we treat LoRA as a strong baseline and focus on **sparse LoRA** methods. While sparse LoRA methods were designed in centralized settings without communication costs in mind, they still serve as reasonable FL baselines. This is because they apply the same principles from sparse backbone methods to LoRA, and to the best of our knowledge, no FL-specific work has proposed applying unstructured sparsity to LoRA.

**FLASC.** In Table 1, we show that the two sparse LoRA baselines can further reduce the communication of LoRA. However, these pruning-based methods apply sparsity to both communication and local finetuning, which is excessively restrictive. Since our goal is to reduce communication, **FLASC** (Algorithm 1

---

**Algorithm 1: FLASC**

1. Require: $r, d_{\text{down}}, d_{\text{up}}, \eta_{\text{s}}, \eta_{\text{c}}, S_{\text{server}}, S_{\text{client}}$
2. $P \leftarrow$ Initialize LoRA with rank $r$
3. $\texttt{optim} \leftarrow \texttt{torch.nn.optim.Adam}(P, \eta_{\text{s}})$
4. **for** $s_{server} = 1, ..., S_{server}$ **do**
5.      $M_{\text{down}} \leftarrow \texttt{TopKMask}(|P|, d_{\text{down}})$
6.      Sample clients $c_1, ..., c_n$ uniformly
7.      at random without replacement
8.      # communication and local training
9.      **for** $i = 1, ..., n$ *in parallel* **do**
10.          $P_i = P \odot M_{\text{down}}$
11.          $P_i' \leftarrow P_i$
12.          **for** $s_{client} = 1, ..., S_{client}$ **do**
13.              $(x, y) \leftarrow$ sample data from $c_i$
14.              $\nabla_{P_i'} = \nabla_1 \mathcal{L}(P_i', x, y)$
15.              $P_i' \leftarrow P_i' - \eta_{\text{client}} \nabla_{P_i'}$
16.          $\Delta P_i \leftarrow P_i - P_i'$
17.          $M_{\text{up},i} \leftarrow \texttt{TopKMask}(|\Delta P_i|, d_{\text{up}})$
18.          $\Delta P_i \leftarrow \Delta P_i \odot M_{\text{up},i}$
19.      # server takes one step of FedAdam
20.      $\texttt{optim.grad} \leftarrow \frac{1}{n} \sum_{i=1}^n \Delta P_i$
21.      $\texttt{optim.step()}$

---

and Figure 2) relaxes the finetuning constraint and only applies sparsity to communication. This simple change enables **FLASC** to match the performance of LoRA while reducing communication. The key difference between **FLASC** and pruning methods lies in the finetuning step; we finetune all entries of $P \odot M$ (L11) and only sparsify the update $\Delta P$ during upload (L17-18). This allows for much more expressive local updates compared to only finetuning the sparse non-zero entries of $P \odot M$. Furthermore, while other baselines propose complex methods for selecting the mask $M$, **FLASC** uses a simple TopK operation with $\ell_1$ criterion.

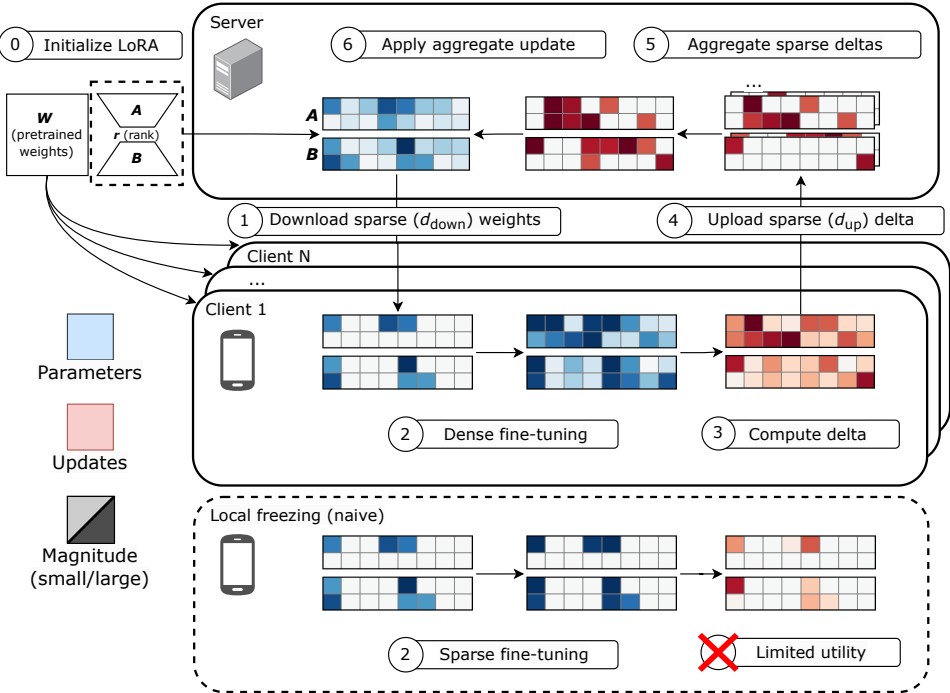

**Figure 2:** A step-by-step overview of **FLASC** with LoRA rank = 2 and density = 1/4. Step 0 is executed prior to FL. Each round of FL repeats steps 1-6. Blue/red squares indicate the magnitude of weights/updates respectively. **Darker** squares indicate a **larger magnitude**, which is the criterion ($\ell_1$) used for sparsity.

**FLASC is as compute-efficient as LoRA. FLASC** finetunes the same parameters as standard LoRA, resulting in the same compute costs. While sparse LoRA methods aim to reduce these costs, their actual compute and memory savings are minimal, since the majority of overhead lies in the backbone's weights and activations and not the LoRA adapters (Cai et al., 2020; Kim et al., 2023a). We analyze these costs in Appendix A.2. Additionally, LoRA can be merged with the backbone after training, which removes its additional inference costs (Luo et al., 2023). Finally, the compute benefits of sparsity can only be realized in specific settings with a high sparsity or structured sparsity (Muralidharan, 2023). Due to these limitations, **FLASC** focuses on using sparsity to reduce the communication of LoRA in FL.

## 4 Results

In this section, we test **FLASC** in a variety of FL settings and show that it is effective at handling concerns of communication efficiency (4.1), hyperparameter tuning (4.2), heterogeneity, (4.3, 4.4), and privacy (4.5). Overall, we show that LoRA with sufficient hyperparameter tuning is robust to these concerns and that **FLASC** is able to match the performance of LoRA with up to 10× less communication.

| Dataset | #Clients | #Examples | #Classes | Metric |
|---|---|---|---|---|
| CIFAR10 | 500 | 50K | 10 | Accuracy (↑) |
| 20NewsGroups | 350 | 20K | 20 | Accuracy (↑) |
| Reddit (next-token prediction) | 20K | 2M | 50257 | Perplexity (↓) |
| Subreddits (topic classification) | 100 | 35K | 50 | Accuracy (↑) |
| FLAIR | 41K | 345K | 17 | F1 Score (↑) |

**Table 2:** Training partition statistics of the datasets.

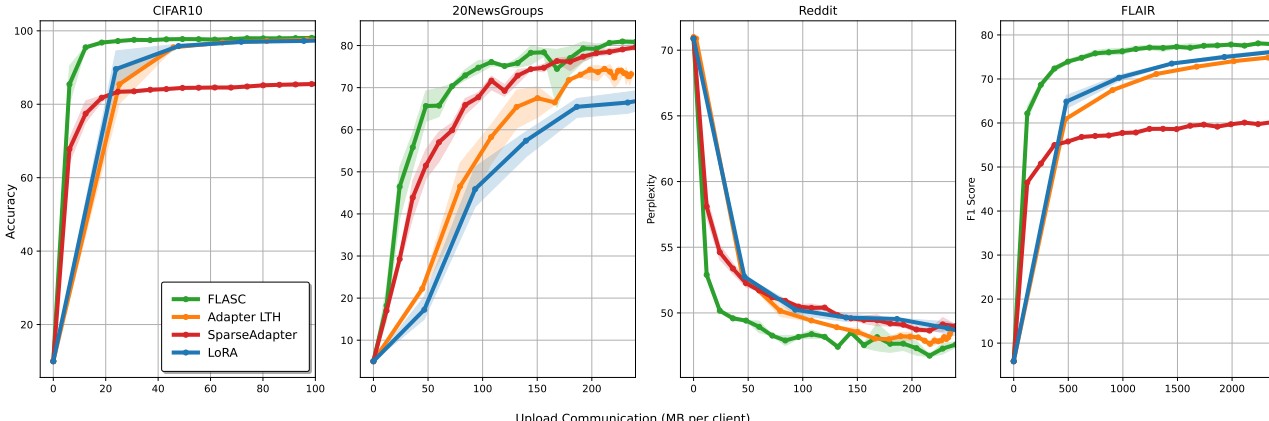

**Figure 3:** We compare utility vs. communication while training LoRA (rank 16) and reducing communication with sparsity. **FLASC** is the most efficient method, while ADAPTER LTH (Wu & Chen, 2022) is inefficient early in training and SPARSEADAPTER (He et al., 2022) may converge to a significantly lower utility than LoRA. Shaded bands show the min/mean/max over 3 random seeds.

**Datasets.** We present experiments on CIFAR10, 20NewsGroups, Reddit, and FLAIR (Krizhevsky, 2009; Lang, 1995; Caldas et al., 2018a; Song et al., 2022) (Table 2). CIFAR10 and FLAIR are image datasets, and 20NewsGroups and Reddit are text datasets. For the image datasets, the images are resized to $224 \times 224$ to match ImageNet (the ViT pretraining dataset). We train with standard data augmentations (random crops and flips). For the text datasets, each example (an email or Reddit comment) is encoded and truncated to a maximum of length of 256 tokens. We use Reddit for two different tasks: the first is a next-token prediction task (Reddit), and the second is a topic classification task (Subreddits). We partition CIFAR10 and 20NewsGroups using a synthetic Dirichlet distribution over the set of labels (Hsu et al., 2019). Reddit and FLAIR are obtained from social media sites (Reddit and Flickr) and are naturally partitioned by user.

**Model Details.** We used ViT-B-16 (85M params) and GPT2-Small (124M params) as the backbone for image and text tasks respectively (Dosovitskiy et al., 2021; Radford et al., 2019). For the image datasets, we finetune a classification head in addition to the LoRA weights. For the text classification tasks (Subreddits and 20NewsGroups), we format the data as a text-to-text task where the model is trained to output the label text (Raffel et al., 2020). For all text datasets, we only finetune the LoRA weights and do not finetune the language modeling head. For all datasets, we use a local batch size of 16. For FLAIR, we sample 200 clients per round, finetune for 2 local epochs, and communicate for up to 5000 rounds. For the other datasets, we sample 10 clients each round, finetune for 1 local epoch, and communicate for up to 400 rounds. Full details on preprocessing and hyperparameters are provided in Appendix B.4.

## 4.1 Communication Efficiency

First, we show each method's utility as a function of communication used during a single training run. In Figure 3, we compare LoRA and the three sparse LoRA methods introduced in Table 1: ADAPTER LTH (Wu & Chen, 2022), SPARSEADAPTER (He et al., 2022), and **FLASC**. We use rank 16 for all methods. The sparsity ratio of ADAPTER LTH (i.e. fraction of parameters removed per round) is tuned separately on each dataset to either 1% or 2%. We use a density of 25% for SPARSEADAPTER and **FLASC**. While we use 25% density across all datasets in this experiment, we show how other methods are unable to handle lower densities in Fig. 5. Across all tasks, **FLASC** matches or exeeds the performance of LoRA while using 3–10× less communication. In contrast, the other two methods cannot reliably match the performance of LoRA. For three out of the four datasets, SPARSEADAPTER converges to a significantly lower accuracy than LoRA. ADAPTER LTH has limited benefits due to its iterative nature; in early rounds, it uses similar communication as LoRA, while in later rounds, the adapters are too sparse to continue training and performance plateaus.

**Communication savings.** In Figure 4, we measure the upload communication that each method needs to reach two target perplexity (PPL) values on Reddit: 53 PPL and 50 PPL. SPARSEADAPTER and ADAPTER LTH reduce per-round communication but take significantly more rounds to reach the same perplexity as LoRA, which leads to limited overall benefits. For example, in the 50 PPL setting, SPARSEADAPTER ends up using the same communication as LoRA, since although it reduces per-round communication by 4×, it also requires 4× as many rounds to match the performance of LoRA. For **FLASC**, we test two upload density values of 1/4 and 1/16. **FLASC** with 1/4 density performs well and does not require any additional rounds to match the utility of LoRA, which results in exactly 1/4 the communication of LoRA. When we further decrease the density to 1/16, **FLASC** requires additional rounds to train, but saves even more communication than 1/4 density. For example, in the 50 PPL setting, **FLASC** with 1/16 density requires $16 * 62/600 \approx 1.65\times$ more rounds than LoRA, but uses $600/62 \approx 10\times$ less upload communication overall.

**Sparsity without freezing.** In Figure 5, we compare **FLASC** to SPARSEADAPTER and FEDERATED SELECT, which are both introduced in Section 3. These two methods respectively correspond to two simple adjustments to **FLASC**: *server-level* and *client-level* freezing. SPARSEADAPTER is an example of server-level freezing. After pruning the model, the zeroed weights are globally frozen; neither the server or clients adjust the mask. FEDERATED SELECT (Charles et al., 2022) is an example of client-level freezing. The server temporarily prunes the model and freezes the zeroed weights during the current round. In other words, clients only download and finetune the non-zero weights. However, unlike SPARSEADAPTER, FEDERATED SELECT adjusts the mask across rounds and can potentially discover more useful weights. While FEDERATED SELECT is proposed in the context of backbone finetuning (and is evaluated as such in Table 1), we apply it to LoRA in Fig. 5. Finally, **FLASC** does not freeze at all; it allows clients to finetune the entire LoRA module and upload sparse updates with varying masks.

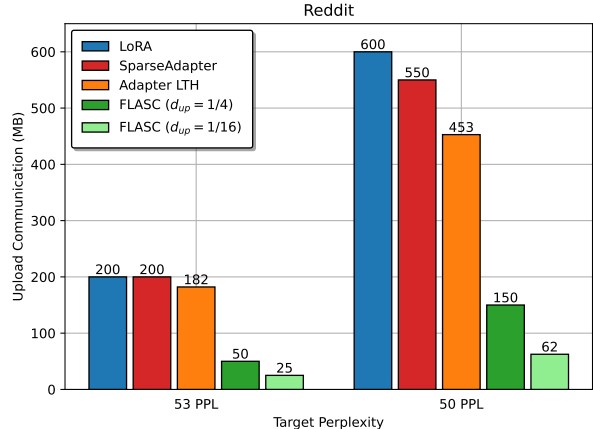

**Figure 4:** We measure upload communication (↓) needed to reach a given perplexity (53 or 50 PPL) on Reddit. **FLASC** (with either 1/4 or 1/16 density) saves significantly more communication than other sparse LoRA baselines.

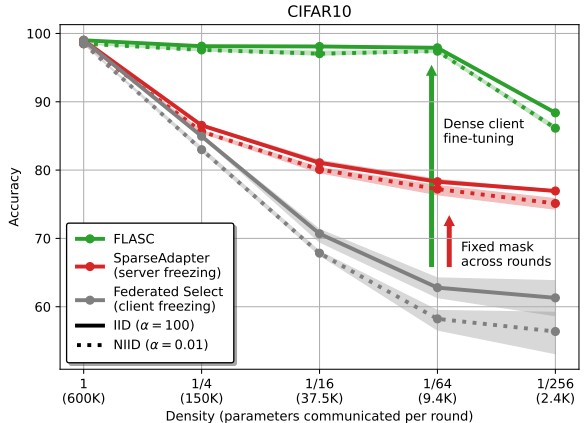

**Figure 5:** We compare **FLASC** to two ways of freezing weights. All methods use the same amount of communication. SPARSEADAPTER and FEDERATED SELECT freeze weights, which significantly harms accuracy. **FLASC** does not freeze weights and only sparsifies communication.

Although freezing generally harms utility, server-level freezing is a simple yet competitive baseline which can outperform more complex methods which dynamically adjust the sparse mask between rounds (Babakniya et al., 2023b). Our results in Figure 5 support this claim, and shows that SPARSEADAPTER works better than FEDERATED SELECT across all density values despite employing a relatively simpler method. Finally, **FLASC** greatly improves utility over both methods by only considering sparse communication without any freezing at all. As discussed in Section 3, freezing LoRA parameters has relatively small compute savings, which motivates us to leverage the utility of dense local updates and then reduce commmunication afterwards.

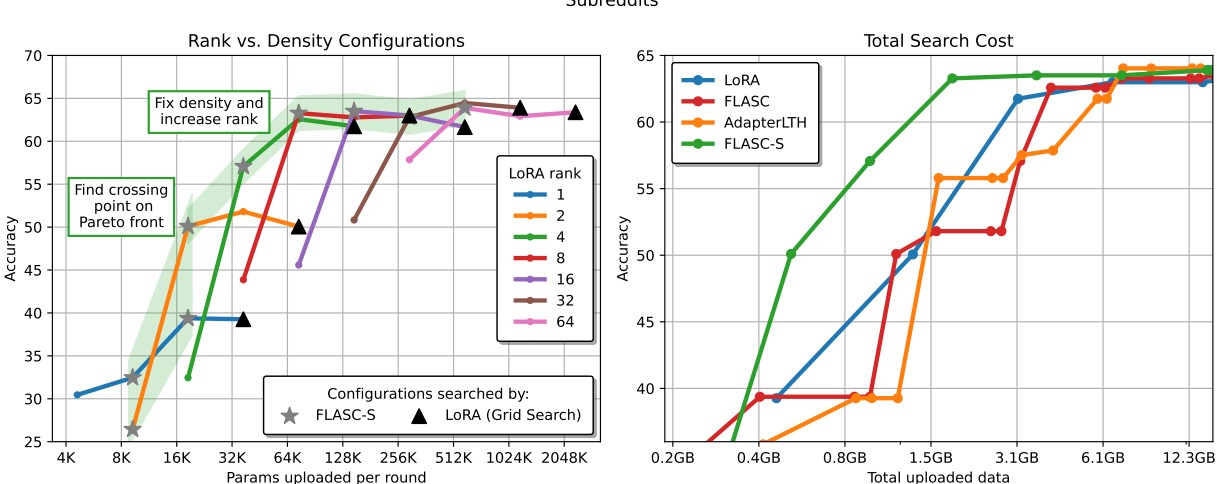

**Figure 6:** We search over a grid of ranks (1 to 64) and densities (1/8 to 1) on the Subreddits task. **Left:** Each curve shows a fixed LoRA rank while varying density. **FLASC-S** efficiently tunes the density on low rank configurations, followed by fixing the density and increasing the rank. **Right:** Sparse LoRA methods (red, orange) introduce additional hyperparameters which are more expensive to tune than the LoRA rank alone (blue). **FLASC-S** (green) efficiently configures **FLASC** by separately tuning density and rank.

## 4.2 Tuning rank and sparsity with FLASC-S

FLASC, like other sparse LoRA methods, requires tuning the rank and sparsity. Generally, a new dataset requires a new search for optimal hyperparameters, and FLASC is no exception. While **FLASC** can train a single model using less communication than LoRA, the additional cost of tuning the sparsity can make **FLASC** more expensive than only tuning the rank of LoRA and therefore negate the benefits of applying sparsity in the first place. To address this issue, we present **FLASC-S** (**FLASC-S**earch), a method for efficiently tuning **FLASC**.

---

**Algorithm 2: FLASC-S**

1 Require: Rank grid $R = \{r_1, ...r_M\}$
2 and density grid $D = \{d_1, ..., d_{N-1}, 1\}$.
3 for $i=1..N$ do
4    $y_1 = \mathbf{FLASC}(r_1, d_i * r_2/r_1)$
5    $y_2 = \mathbf{FLASC}(r_2, d_i)$
6    if $y_1 < y_2$ then
7      | break
8 for $j=i..M$ do
9    $y_j = \mathbf{FLASC}(r_3, d_i * r_1/r_2)$

---

The left plot of Figure 6 provides a high-level illustration of how **FLASC-S** works. We sweep over rank and density values increasing by a factor of 2 and highlight the configurations that are evaluated by **FLASC-S**. A critical observation is that points on the Pareto front apply a similar density ($0.25 - 0.5\times$), while a lower density ($0.125\times$) performs worse than lower rank ($8\times$) configurations of equal cost. Furthermore, too low of a rank harms performance, while too high of a rank results in redundant communication.

Based on these observations, our tuning method **FLASC-S** (Alg. 2.) tunes the sparsity using the lowest-rank configurations, fixes the sparsity, and then increases the rank. Given two equal-cost (rank, density) configurations $(r_2, d)$ and $(r_1, dr_2/r_1)$ where $r_1$ and $r_2$ are the two lowest ranks in the search space, we expect $r_2$ to perform worse when $d$ is extremely small. However, we also expect $r_2$ to improve more quickly than $r_1$ as $d$ is increased. This allows us to sweep over values of the density $d$; once we encounter a value where $r_2$ performs better than $r_1$, we know $(r_2, d)$ lies on the rank-density Pareto front. After finding this configuration, we fix $d$ and increase the rank to find other configurations which lie close to the Pareto front.

The right plot of Figure 6 shows the communication cost of using **FLASC-S** versus running grid search on sparse LoRA baselines. The cost (x) is measured in terms of the accumulated communication from testing multiple (rank, sparsity) configurations, and the performance (y) is given by the best configuration found so far. Using grid search to tune sparse LoRA baselines (including **FLASC**) uses more communication than regular LoRA due to the cost of tuning extra sparsity hyperparameters. By tuning density on the smallest ranks before scaling up the rank, **FLASC-S** provides cheaper tuning costs than LoRA.

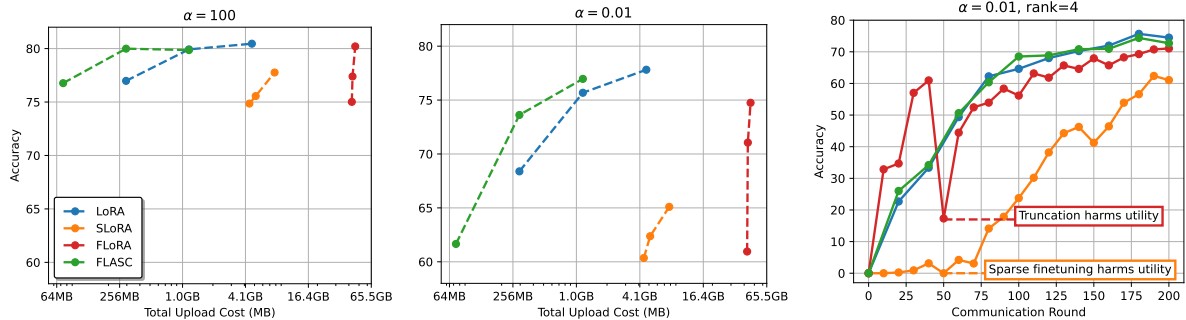

**Figure 7:** We compare **FLASC** to LoRA and SLoRA on 20NewsGroups. **Left, middle**: We test two settings of data heterogeneity $\alpha$. Each curve varies the LoRA rank from 1 to 16. **FLASC** with 0.25 density improves over LoRA in all settings, while FLoRA (SLoRA) has large overhead costs due to an initial 50 rounds of "Stage 1" full-rank finetuning. **Right**: We compare the per-round accuracy of methods. FLoRA performs best during Stage 1, but degrades significantly after low-rank truncation. SLoRA applies a 10% density to Stage 1, which saves communication but harms overall accuracy.

### 4.3 Data Heterogeneity

Data heterogeneity is a commonly studied issue which can harm FL optimization (Li et al., 2020). In the context of LoRA, Babakniya et al. (2023a) finds that LoRA suffers more from data heterogeneity compared to full-rank finetuning and proposes FLoRA (SLoRA), a method that learns a full-rank (and sparse) update and then initializes LoRA with its truncated SVD (see Appendix A).

Our experiment in this section compares **FLASC** to LoRA, FLoRA, and SLoRA under varying levels of data heterogeneity $\alpha$. Following standard practice in FL literature, we partition the data using a Dirichlet distribution over the set of labels (Hsu et al., 2019). Each client draws a vector from a Dirichlet$(\alpha)$ distribution, where the vector dimension is equal to the number of labels. We then normalize each dimension by that label's global frequency, and the resulting vectors correspond to the clients' individual label frequencies. We show two separate plots for the heterogeneity parameter $\alpha \in \{100, 0.01\}$. At $\alpha = 100$, clients have an approximately uniform number of examples per label, while at $\alpha = 0.01$, over 90% of most clients' examples belong to a single label.

In Figure 7, each curve (left and middle plots) shows LoRA with rank varying from $r \in \{1, 4, 16\}$. A 4× larger rank costs 4× more total upload (x-axis). When the data is identically distributed ($\alpha = 100$), LoRA rank 1 performs well and achieves only 1-2% less accuracy than larger ranks. When the data is heterogeneous ($\alpha = 0.01$), all ranks perform worse, but the impact on lower ranks is much more significant; the accuracy gap between LoRA rank 1 versus rank 16 grows from 2 to 12%. As long as we select an adequately large rank, **FLASC** is a strong baseline in both i.i.d. and non-i.i.d. settings. **FLASC** can use a density of 0.25× with minor impact on accuracy, resulting in $2 - 8\%$ higher accuracy compared to LoRA with a 4× smaller rank.

Finally, we evaluate FLoRA, which runs an initial 50 rounds of full-rank finetuning called "Stage 1" (see Appendix A). Unfortunately, full-rank finetuning costs over 200× the communication of LoRA and dominates the overall cost of finetuning (left and middle plots). A modification to reduce this cost called SLoRA applies sparsity to Stage 1, but this severely limits model utility (right plot). Furthermore, SVD truncation after Stage 1 severely harms accuracy and is worse than simply using LoRA throughout. Therefore, while FLoRA improves quickly and performs best on a per-round basis during Stage 1, these benefits cannot be fully realized due to (1) high communication cost (up to 200× the cost of LoRA) and (2) performance drop after SVD truncation.

### 4.4 Systems Heterogeneity

In addition to data heterogeneity, FL settings also face issues of systems heterogeneity where clients with limited system resources can slow down or harm training. In recent work, Cho et al. (2023) develop HETLORA,

| Method (Description) | Accuracy | | | |
|---|---|---|---|---|
| | $\alpha = 1$ | | $\alpha = 0.01$ | |
| | 3 tiers | 5 tiers | 3 tiers | 5 tiers |
| All lowest tier (rank 1) (Restrict all clients to lowest communication tier) | 77.4 | | 68.4 | |
| Highest tier only (rank 16) (Drop lower-tier stragglers from training) | 74.6 | 72.0 | 56.2 | 45.1 |
| HetLoRA (Clients train LoRA with varying rank) | 77.8 | 76.8 | 64.3 | 40.0 |
| **FLASC** (Clients upload sparse LoRA updates) | **80.7** | **80.4** | **75.8** | **78.1** |

**Table 3:** We compare methods for handling systems heterogeneity on 20NewsGroups. We vary both data and systems heterogeneity in terms of the label frequency ($\alpha$) and tiers of upload budget (tiers) respectively. Data heterogeneity exacerbates the issues of systems heterogeneity. **FLASC** is more robust than other baselines to both data and systems heterogeneity.

which aims to address systems heterogeneity by training LoRA modules with different ranks across clients. During training, client $c$ with assigned rank $r_c$ will download the uppermost $r_c$ rows of $A$ and leftmost $r_c$ columns of $B$ from the global LoRA weights (with rank $r_s$) and then use the weights to initialize a local LoRA module (with rank $r_c$). For a fair comparison with other methods in terms of hyperparameter tuning, we do not use the self-pruning method proposed by HETLoRA (i.e. we set $\gamma = 1, \lambda = 0$).

Our following experiment on systems heterogeneity constrains the upload budget of each client. We consider two heterogeneity settings of low ($b_s = 3, r_{\text{base}} = 4$) and high ($b_s = 5, r_{\text{base}} = 2$), where $b_s$ is the number of **tiers (unique upload budgets)** and $r_{\text{base}}$ is the **growth factor between budget tiers**. Each client $c$ is assigned to one of these upload budgets $b_c \in \{1, 2, ..., b_s\}$ uniformly at random. To satisfy the upload budget, HETLoRA assigns clients a local rank $r_c = r_{base}^{b_c-1}$. **FLASC** uses the same upload communication as HETLoRA by finetuning a rank $r_s$ module and applying an upload density of $d = r_c/r_s = r_{\text{base}}^{(b_c-b_s)}$. For both methods, the server initializes a LoRA adapter with rank $r_s = r_{\text{base}}^{b_s-1} = 16$. In addition, we evaluate two baselines **"All lowest tier"** and **"Highest tier only"**. To address settings where clients have significantly different upload speeds, "All lowest tier" restricts all clients to LoRA rank 1 in order to accommodate the clients with the slowest upload speeds. On the other hand, "Highest tier only" drops the clients with slow upload and only uses model updates from clients which are assigned the highest budget (rank 16).

In Table 3, we show that **FLASC** outperforms HETLoRA as well as the two naïve baselines. Generally, performance is better when systems are less (3 tiers) as opposed to more (5 tiers) heterogeneous. Additionally, we vary the data heterogeneity between $\alpha = 1$ or 0.01 (see Section 4.3). Heterogeneous data ($\alpha = 0.01$) can significantly exacerbate issues of systems heterogeneity; this is because data heterogeneity affects the accuracy of LoRA at lower ranks, while systems heterogeneity affects which clients are assigned to those lower ranks. When $\alpha = 0.01$, "Highest tier only" performs extremely poorly because it can only utilize a small set of skewed training data. Surprisingly, HETLoRA performs worse than "Highest tier only" when $\alpha = 0.01$, tiers = 5. In this setting, HETLoRA performed better in earlier rounds (due to leveraging data from more clients), but converged to a worse solution (due to heterogeneous updates of varying rank).

## 4.5 Privacy

Finally, we consider the performance of **FLASC** when used in conjunction with differential privacy. FL model updates are susceptible to privacy leakage and adversarial attacks (Geiping et al., 2020), which necessitates additional techniques such as differential privacy (DP). DP is a popular framework that adds randomness to an algorithm in order to mask example-level contributions to the algorithm's output (Abadi et al., 2016; McMahan et al., 2017b). Full finetuning scales poorly to strict DP budgets due to DP noise overwhelming the signal in the model updates, but PEFT (e.g. LoRA) is a promising solution to this problem (Luo et al.,

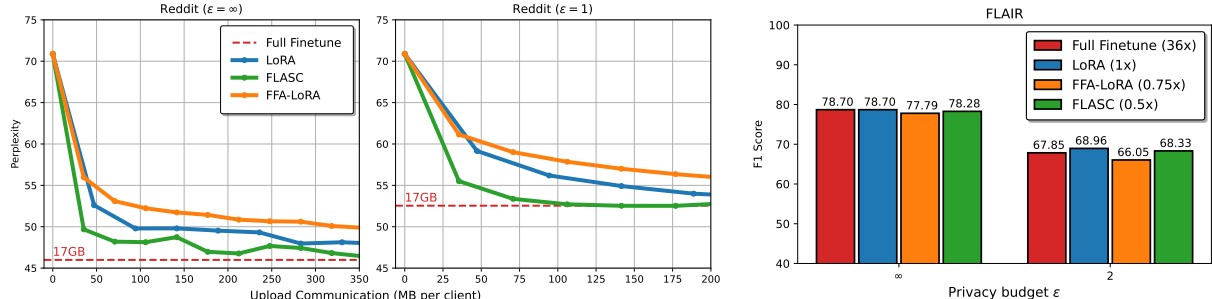

**Figure 8:** We compare finetuning methods with DP-FEDADAM. While full finetuning performs best in non-private settings, LoRA is much cheaper and can even outperform full finetuning in private settings. FFA-LoRA harms utility and is less efficient than LoRA, while **FLASC** improves the efficiency of LoRA.

2021; Yu et al., 2022). In the FL space, Sun et al. (2024) suggest that combining FL and LoRA can amplify noise from DP and proposes FFA-LoRA, a method which freezes the $A$ matrix of LoRA.

Our following experiments evaluate FL methods in non-private ($\epsilon = \infty$) and private ($\epsilon < \infty$) settings. $\epsilon$ is a privacy budget parameter which bounds the sensitivity of the algorithm's output with respect to any individual client in the training data. We compare full finetuning, LoRA, FFA-LoRA (reduce comm. by 25%), and **FLASC** (reduce comm. by 75% on Reddit, 50% on FLAIR). In Figure 8, we show the communication efficiency (left) and final utility (right) of each method. Similar to prior works in DP, we find that LoRA is more robust to DP than full finetuning. While FFA-LoRA reduces communication, it also significantly harms utility and leads to worse efficiency. In Figure 9, we further study the effect of parameter count by varying the rank of LoRA from 1 to 64. We find that a larger rank does better in non-private settings, while the optimal rank is smaller in private settings. Overall, using **FLASC** with a sufficiently small rank is both more accurate and efficient than applying FFA-LoRA to any rank.

## 5 Conclusion

In this paper, we introduce **FLASC**, an efficient FL method that significantly reduces the communication cost of LoRA. Our method provides both higher utility and substantial communication savings relative to existing pruning-based methods. Furthermore, we show that methods which introduce sparsity to LoRA can generally increase communication costs once accounting for hyperparameter tuning, due to the

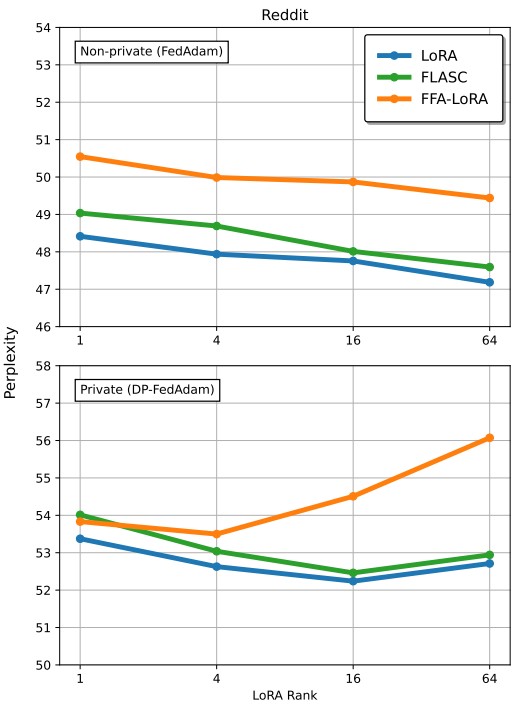

**Figure 9:** We vary the rank of LoRA in private vs non-private settings. Higher rank is generally better in non-private settings, while the optimal rank is lower in private settings.

extra hyperparameters these methods introduce. To address this issue, we propose **FLASC-S**, a method which sequentially tunes the sparsity of **FLASC** followed by the rank. Finally, we find that **FLASC** is competitive with specific solutions for other FL concerns of heterogeneity and privacy while achieving superior communication efficiency. Overall, our results indicate that **FLASC** can serve as a strong baseline for future works in federated fine-tuning. Still, many important questions remain on how to make LoRA even more efficient in resource-constrained federated networks. For instance, LoRA itself may be insufficient when scaling to even larger models. In the future, we aim to investigate such questions and design methods to make high-quality models more accessible to low-resource users.

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

# A    Methods

## A.1    FL Optimization with Sparsity and LoRA.

**FedAdam.** FEDADAM is an optimization method that follows the FedOpt framework (Reddi et al., 2020). At every round, each participating client $i$ downloads a copy of the trainable weights $P$, finetunes $P$ to obtain updated weights $P_i'$, and uploads $\Delta P_i = P - P_i'$ to the server. The server then computes an average update $\Delta P = \frac{1}{n}\sum_{i=1}^n \Delta P_i$, where $n$ is the number of clients sampled per round. The average may optionally be weighted by each client's dataset size. $\Delta P$ can be interpreted as a global pseudo-gradient; for example, the update rule for FedAvg is to set $P \leftarrow P - \Delta P$ for the next round (McMahan et al., 2017a). In the case of FEDADAM, a stateful Adam optimizer (Kingma, 2014) takes $\Delta W$ as input and outputs an adapted global update at each round. In Algorithm 3, we show how Adam and FedAdam differ only in how $\Delta P$ is computed.

We use a single parameter vector $P$ to refer to the trainable parameters of the model. For LoRA, $P$ is a flattened and concatenated vector of LoRA weights $\{A_l, B_l\}_{l=1}^L$ where $L$ is the number of layers LoRA is applied to. Applying `TopKMask` to $P$ naturally results in varying sparsity per layer. An alternative approach is to uniformly sparsify each layer $(A_l, B_l)$ in before concatenation, but the former tended to perform better.

---

**Algorithm 3:** General functions for central and federated training with pruning

1 # FL-specific lines are colored red.
2 **Require:**
3     $P$ (trainable parameters)
4     $\mathcal{L}$ (loss function)
5     $S_{\text{server}}$ (central training steps or FL rounds)
6     $\eta_{\text{server}}$ (learning rate)
7     $M$ (sparse mask)
8     $S_{\text{client}}$ (num. local steps)
9     $\eta_{\text{client}}$ (client learning rate)
10    $n$ (clients per round)
11 **Function** `FLRound`$(P, M, S_{\text{client}}, \eta_{\text{client}}, n)$**:**
12     Sample client datasets $c_1, ..., c_n$ uniformly at random w/o replacement
13     **for** $i = 1, ..., n$ in parallel **do**
14         $P_i \leftarrow P$
15         **for** $s = 1, ..., S_l$ **do**
16             Sample batch of data $(x, y)$ from $c_i$
17             $\nabla_{P_i} = \nabla_1 \mathcal{L}(P_i, x, y) \odot M$ # sparse gradient
18             $P_i \leftarrow P_i - \eta_{\text{client}} \nabla_{P_i}$
19         $\Delta P_i \leftarrow P_i - P$
20     return $\frac{1}{n}\sum_{i=1}^n \Delta P_i$
21 **Function** `TrainStep`$(P, M)$**:**
22     **if** FEDADAM **then**
23         return `FLRound`$(P, M, S_{client}, \eta_{client}, n)$
24     **else if** ADAM **then**
25         Sample batch of data $(x, y)$ from central dataset
26         return $\nabla_1 \mathcal{L}(P, x, y) \odot M$ # sparse gradient

---

**Algorithm 4:** Standard LoRA training

1 $P \leftarrow$ Initialize LoRA with rank $r$
2 $M \leftarrow 1$ # all-ones matrix
3 `optim` $\leftarrow$ `torch.nn.optim.Adam`$(P, \eta_{\text{server}})$
4 **for** $s_{\text{server}} = 1, ..., S_{\text{server}}$ **do**
5     $P.\text{grad} \leftarrow$ `TrainStep`$(P, M)$
6     `optim.step()`

**Sparse LoRA and Sparse FL baselines.** In addition to showing how Adam and FedAdam both fit in the same general training framework, Algorithm 3 shows how centralized pruning methods can be adapted to FL. We use this framework to succinctly describe our baselines i.e. LoRA in Alg. 4, iterative pruning methods ADAPTER LTH and FEDSPARSIFY-GLOBAL in Alg. 5, and fixed density methods SPARSEADAPTER, SPDST, and FEDERATED SELECT in Alg. 6.

**Iterative Magnitude Pruning.** We use iterative magnitude pruning (IMP) without weight rewinding. In the original Lottery Ticket Hypothesis (Frankle & Carbin, 2018), the authors propose using IMP to find sparse subnetworks that can be trained starting from their randomly initialized weights. However, the procedure of identifying this subnetwork is extremely expensive, as it requires $10-30$ iterations of (1) training the model to convergence, (2) removing the $\%p$ smallest magnitude weights, and (3) rewinding (i.e. resetting) the remaining weights to their initial values. Therefore, this method is unsuitable if the goal is simply to efficiently train a single model. Therefore, Renda et al. (2019) proposes a more efficient alternative which is to skip the rewinding step and instead continue training from the pruned state. They show that "IMP without rewinding" is also able to identify trainable sparse subnetworks without the expensive cost of rewinding.

---

**Algorithm 5:** Iterative Magnitude Pruning (IMP) without rewinding

**1 Require:** $p$ (prune ratio), $q$ (prune frequency)
**2 if** ADAPTER LTH **then**
**3** | $P \leftarrow$ Initialize LoRA with rank $r$
**4 else if** FEDSPARSIFY-GLOBAL **then**
**5** | $P \leftarrow$ Initialize model backbone weights
**6** $d \leftarrow 1$
**7** optim $\leftarrow$ `torch.nn.optim.Adam`$(P, \eta_{\text{server}})$
**8 for** $s_{\text{server}} = 1, ..., S_{\text{server}}$ **do**
**9** | **if** $0 \equiv s_{server} \mod q$ **then**
**10** | | $M \leftarrow$ `TopKMask`$(|P|, d)$
**11** | | $P \leftarrow P \odot M$ # permanently remove entries in $P$
**12** | | $d \leftarrow d(1-p)$
**13** | $P$.`grad` $\leftarrow$ `TrainStep`$(P,M)$
**14** | optim.step()

---

**Pruning-at-initialization.** Pruning-at-initialization (PaI) is a class of methods which prune the model to a fixed density at the start of training. While IMP (as its name suggests) generally uses magnitude to rank parameters, pruning-at-initialization (PaI) methods use various sensitivity metrics which determine the parameters to prune. In our work, we consider SPARSEADAPTER, which applies PaI to LoRA in centralized learning, and SPDST, which applies PaI to full finetuning in federated learning.

**SparseAdapter.** As discussed in Section 3, SPARSEADAPTER as proposed is unsuitable for pruning of LoRA. In general, any senstivity metric which depends on parameter magnitude is unsuitable for LoRA because the $B$ matrix in LoRA is initialized to all zeros. Therefore, all $B$ weights will be pruned before any weights in $A$. For matrices $B \in \mathbb{R}^{d \times r}, A \in \mathbb{R}^{r \times k}$, pruning at any density below $d/(d+k)$ will entirely prune the $B$ matrix, which leaves us unable to train the model. Therefore, instead of using the SNIP metric, we use parameter magnitude after finetuning the model for a limited number of iterations.

**SPDST.** SPDST locally runs a pruning method called SNFS (Dettmers & Zettlemoyer, 2019) and produces a sparse model on each client. It then measures the average density of each layer across clients to determine which layers are more or less sensitive. Finally, it randomly prunes each layer of the initial model proportionally to its sensitivity. We found that SPDST has limited utility due to (1) producing sparse models with SNFS and (2) randomly selecting the final sparse parameters. Therefore, we evaluate a simpler pruning-at-initialization metric which first runs a single round of full finetuning and aggregation, then applies magnitude pruning to the aggregate model. These two adjustments improve performance while sticking to the same principle as SPDST, which is to directly finetune the model in order to determine which parameters should be pruned.

**Federated Select.** FEDERATED SELECT is a method which falls under the broader category of fixed-density pruning. Like the two PaI methods above, FEDERATED SELECT prunes the model to a given density and

freezes the zeroed weights during client finetuning. However, unlike PaI methods, FEDERATED SELECT retains the original dense weights at the server; after aggregation, it can adjust the mask based on the updated weights and have clients download a model with an updated sparsity pattern in the following round.

---

**Algorithm 6:** Fixed-density pruning

**1**   **Require:** $d$ (prune density)
**2**   **if** SPARSEADAPTER **then**
**3**      $P \leftarrow$ Initialize LoRA with rank $r$
**4**   **else**
**5**      # We use FEDERATED SELECT to prune the backbone in Table 1 and to prune LoRA in Sec. 4.2.
**6**      $P \leftarrow$ Initialize model backbone weights
**7**   # Compute importance scores; SPARSEADAPTER proposes using SNIP (gradient-magnitude) sensitivity, but this is unsuitable for LoRA. (see Sec. 3). Instead, we use "SPDST" sensitivity.
**8**   **if** SPDST **then**
**9**      # We use a modified version of SPDST which finetunes the model and computes TopK weights.
**10**     $M \leftarrow 1$ # matrix of all ones
**11**     $P' \leftarrow$ TrainStep($P$,$M$)
**12**     $S \leftarrow |P'|$
**13**     $M \leftarrow$ TopKMask($S, d$)
**14**   **if** not FEDERATED SELECT **then**
**15**     $P \leftarrow P \odot M$ # permanently remove entries in $P$
**16**   optim $\leftarrow$ torch.nn.optim.Adam($P, \eta_{\text{server}}$)
**17**   **for** $s_{\text{server}} = 1, ..., S_{\text{server}}$ **do**
**18**     **if** FEDERATED SELECT **then**
**19**       $M \leftarrow$ TopKMask($|P|, d$)
**20**       $P' \leftarrow P \odot M$ # download a temporarily sparse $P'$
**21**     **else**
**22**       $P' = P$ # $P$ is already sparse
**23**     $P$.grad $\leftarrow$ TrainStep($P'$,$M$)
**24**     optim.step()

---

**FedSpa.** FEDSPA is a sparse backbone method proposed by Huang et al. (2022). Similar to FEDERATED SELECT, the server maintains a dense global model while clients download a sparse model and only finetune the non-zero parameters of these sparse models. FEDSPA additionally learns a personalized mask at each client. After local finetuning, FEDSPA runs a local "mask searching" algorithm which adjusts the mask in advance of the next round the client is selected for training. Unfortunately, this introduces a limitation in that the personalized masks must be statefully maintained across rounds, which is expensive when a large number of clients participate in FL. We found that FEDSPA performed worse than other sparse backbone methods under a limited budget of 200 rounds, though it can achieve non-trivial performance with a larger budget of rounds.

| Method / Round | 200 | 400 | 600 |
|---|---|---|---|
| FEDSPA (density=0.75) | 5.00 | 49.48 | 5.00 |
| FEDSPA (density=0.5) | 5.00 | 16.67 | 40.17 |

**Table 4:** Evaluation of FEDSPA on 20NewsGroups. While FEDSPA attains non-trivial accuracy after 400 rounds, it performs worse than full finetuning and LoRA, which both reach 78% accuracy in 200 rounds.

## A.2   Sparsity details.

**Communication cost of the mask.** The mask is stored as a binary vector with $p$ entries where $p$ is the number of trainable parameters in the model. The cost of communicating this mask is $p$ bits, which is independent of the actual amount of sparsity in the mask. We use this format for simplicity and because **FLASC** (like other model sparsity methods) requires relatively large density values. If communication is sufficiently sparse, it would be beneficial to use a sparse matrix format which scales with sparsity. Such a matrix format (e.g. CSR) stores only the non-zero indices but each index (e.g. uint8) requires multiple bits.

**Sparsity pattern. FLASC** uses an unstructured mask where each bit corresponds to a single parameter. For LoRA adapters of dimension $d \times r$ and $r \times k$, an unstructured mask stores $r(d+k)$ bits, while a structured (feature-level) mask only stores $d + k$ bits which correspond to the rows of $B$ and columns of $A$. While FLASC can be extended to structured (i.e. feature-level) sparsity, unstructured sparsity is more expressive and results in better utility. To support this, we ran an experiment which shows that (1) structured sparsity saves only a little total communication and (2) harms model training. Structured sparsity can be effective if some features are naturally sparse over all the data of a client, but this is unlikely in practice.

| Mask Structure | 20NewsGroups (↑) | Reddit (↓) | Parameter size | Mask size | Total upload |
|---|---|---|---|---|---|
| Unstructured (weights) | **81.4** | **46.7** | 0.6 MB | 0.0750 MB | 270 MB |
| Structured (features) | 64.4 | 47.9 | | 0.0047 MB | 242 MB |

**Table 5:** Comparison of applying different sparsity structures to **FLASC** with rank 16 and 0.25 density. Using structured sparsity reduces the storage cost of the mask, but significantly harms performance. Total upload (per client) is the parameter size + mask size, multiplied by 400 FL rounds.

**Sparsity criteria.** For practical reasons, we choose the $\ell_1$ norm of the delta (local weight update) over more complex importance metrics such as historical information across rounds (which is not available in stateless cross-device settings), or gradients (which are already incorporated during finetuning). The theoretical justification is that our goal is to sparsify the model update while remaining close to the updated model, which makes the delta $\ell_1$ the most natural choice. To validate this intuition, we ran an experiment which shows that delta $\ell_1$ performs better than two more complex baselines: gradient and gradient-delta product.

| $\ell_1$ Criteria | 20NewsGroups (↑) | Reddit (↓) |
|---|---|---|
| Delta | **81.4** | **46.7** |
| Gradient | 78.9 | 47.2 |
| Delta-Gradient Product | 80.6 | 46.9 |

**Table 6:** Comparison of $\ell_1$ mask criteria. We report the best performance over 400 FL rounds.

**Compute benefit of sparsity.** As discussed in Section 3, prior sparse LoRA works have marginal benefits because most of the computation occurs in the model backbone. Therefore, a key takeaway of our work is that combining sparsity with LoRA is uniquely beneficial for reducing communication. In Table 7, we profile the FLOPs used by various model sizes during one forward-backward pass on a 128-token input sequence.

| Model | Parameters | | | GFLOPs | | | Ratio vs. Base | |
|---|---|---|---|---|---|---|---|---|
| | Base | $r = 1$ | $r = 64$ | Base | $r = 1$ | $r = 64$ | $r = 1$ | $r = 64$ |
| GPT2-Small | 124M | 37K | 2.4M | 62.86 | 0.04 | 1.81 | 0.07% | 2.88% |
| GPT2-Medium | 355M | 98K | 6.3M | 180.33 | 0.11 | 4.84 | 0.05% | 2.69% |
| GPT2-Large | 812M | 184K | 11.8M | 394.39 | 0.20 | 9.10 | 0.05% | 2.31% |

**Table 7:** We use the PyTorch profiler to compare the cost of the base model vs. LoRA adapters of rank $r$. "GFLOPs" is $10^9 \times$ the number of floating-point operations involving base model or LoRA parameters.

### A.3 FL LoRA methods.

**SLoRA.** FLoRA and SLoRA (Babakniya et al., 2023a) are two-stage methods that run multiple rounds of full-rank finetuning (Stage 1), apply SVD decomposition and truncation to the model update, and then train LoRA using the truncated weights as an initialization (Stage 2). The motivation is that when data in FL is more heterogeneous, a higher effective rank is beneficial for capturing the diversity in client updates. Unfortunately, full-rank finetuning (as used in FLoRA) can be expensive and dominates communication costs (see Sec. 4.3). To reduce this cost, SLoRA is a modification that applies weight-level sparsity i.e. 90% of the full-rank parameters are frozen at the start of Stage 1.

**FFA-LoRA.** FFA-LoRA (Sun et al., 2024) is a method which freezes the $A$ matrix of LoRA. The communication savings of FFA-LoRA depends on the relative size of the $A, B$ matrices, which varies across layers and model architectures. In our experiments, we apply LoRA to the K,V (CIFAR10 / FLAIR) or K,V,Q (20NewsGroups, Reddit) projections in the attention layers. For these two settings, FFA-LoRA uses 2/3 and 3/4 of the original communication respectively. This savings is smaller than 1/4 density we typically apply to **FLASC**.

We present an additional experiment which shows that the optimization method can affect the performance of FFA-LoRA vs. LoRA. The original FFA-LoRA work uses FEDAVG, while our work uses FEDADAM. We chose FEDADAM because it is a state-of-the-art FL optimizer which is commonly used in practice and generally outperforms FEDAVG. With FEDAVG, FFA-LoRA slightly improves over LoRA when using a large batch size and significantly improves when using a small batch size. We hypothesize that LoRA can fail to train in a small-batch setting due to noise from stochastic gradients compounding with the inexact aggregation of LoRA. However, FEDADAM appears to mitigate these issues. When using FEDADAM with a small batch size (16), both LoRA and FFA-LoRA outperform their FEDAVG counterparts and LoRA performs better than FFA-LoRA.

| Method | FEDAVG, Batch=200 | FEDAVG, Batch=16 | FEDADAM, Batch=16 |
|---|---|---|---|
| LoRA | 91.45 | 50.00 | **93.65** |
| FFA-LoRA | **91.74** | **90.77** | 92.68 |

**Table 8:** Comparison of FFA-LoRA vs. LoRA on a 3-client IID partition of QNLI. FFA-LoRA performs better with FEDAVG, but LoRA performs better with FEDADAM. Additionally, FEDADAM generally outperforms FEDAVG and can tolerate a smaller batch size.

FFA-LoRA is designed for the stronger notion of *local* (as opposed to *global*) differential privacy. In local DP, clients locally run DP-SGD, which is only feasible in *cross-silo* FL settings where each client has a large number of local examples. In contrast, we focus on cross-device settings which assume a large widespread pool of clients with few examples per client.

**Differentially Private (DP)-FedAdam.** To apply global DP to FEDADAM, clients upload updates computed by non-private SGD. The server clips these updates, aggregates them, normalizes by the clipping norm, and then adds Gaussian noise with scale $\sigma$ (De et al., 2022). This protects client privacy at a coarse-grained level where the "neighboring datasets" definition of DP applies to the addition or removal of one client's local dataset rather than a single example (McMahan et al., 2017b).

To obtain reasonable privacy guarantees when using DP-FEDADAM, we must bound the sensitivity of the aggregate update with respect to any individual client. The most obvious way to achieve this is to sample a large cohort of clients (Charles et al., 2021). However, when running experiments with private FL, this can make training costs prohibitively expensive. To make simulation feasible in terms of wall-clock time, a common trick is to select a large ('simulated') client cohort size, compute the noise scale according to the privacy constraints, and then linearly scale it down according to a smaller cohort size actually used for experiments. For instance, Song et al. (2022) (Sec. 5.1, p.7) uses "200 users sampled per round to simulate the noise-level with a cohort size of 5,000". We follow this simulation setup for our experiments on FLAIR. For Reddit, we sample 10 users per round and simulate the noise-level with a cohort size of 1,000. Simulating a larger cohort size has two effects: (1) our models are less private than the reported privacy budget, but (2)

provide a *lower bound* on the accuracy we would obtain from training with the true cohort size / privacy budget.

# B Experiment Details

| Dataset | Task | Partition | #Clients | #Examples | #Classes |
|---|---|---|---|---|---|
| CIFAR10 | Image Classification | Dirichlet | 500 | 50K | 10 |
| 20NewsGroups | Sequence Classification | Dirichlet | 350 | 20K | 20 |
| Reddit | Next Token Prediction | Natural | 32K | 1.1M | 50257 |
| Subreddits | Sequence Classification | Natural | 100 | 35K | 50 |
| FLAIR | Object Detection | Natural | 41K | 345K | 17 (multilabel) |

**Table 9:** More detailed information on the datasets used in the experiments.

## B.1 Artifacts

We provide the code in the supplementary material of our submission.

## B.2 Compute Resources

We simulate FL training on a single NVIDIA L40S GPU and parallelize trials over multiple GPUs.

## B.3 Training Details

We apply LoRA to the attention layers only, specifically the K,V mappings for ViT, and K,V,Q mappings for GPT2. For CIFAR10 and FLAIR, we train a classification layer and the LoRA parameters. On 20NewsGroups and Subreddits, we do not train a classification layer. Instead, we format the dataset such that the model outputs the label in text form. We freeze the language modeling (embedding) head and only finetune LoRA parameters. Reddit is a next-token prediction task so we do not perform any additional modification to the model or data.

## B.4 Hyperparameters

FEDADAM hyperparameters:

- Learning rates $\eta_{\text{server}}, \eta_{\text{client}}$:
  - CIFAR10: 5e-3, 1e-3
  - 20NewsGroups: 1e-2, 1e-3
  - Reddit: 1e-2, 1e-3
  - FLAIR: 1e-3, 1e-2

- Betas: $\beta_1 = 0.9, \beta_2 = 0.999$

- Clients per round: 10 (CIFAR10, 20Newsgroups, Reddit), 200 (FLAIR)

- Training rounds:
  - CIFAR10: 200
  - 20NewsGroups: 200 (Fig. 1, 7, Tab. 3), 400 (Fig. 3)
  - Reddit: 200 (Fig. 6), 400 (Fig. 4, 8, 9)
  - FLAIR: 5000

- Client optimizer: SGD (batch size= 16, momentum= 0.9)

Figure 1:

1. 1 GB budget:
   - LoRA rank $r$: 4
   - **FLASC**, ADAPTER LTH, SPARSEADAPTER rank $r$: 16
   - **FLASC**, SPARSEADAPTER density: 0.25
   - ADAPTER LTH density: 0.98
   - Training rounds: 200

2. 128 MB budget:
   (a) LoRA rank $r$: 1
   (b) **FLASC**, ADAPTER LTH, SPARSEADAPTER rank $r$: 4
   (c) **FLASC**, SPARSEADAPTER density: 0.25
   (d) ADAPTER LTH density: 0.96
   (e) Training rounds: 100

Figure 3:

- LoRA rank $r$: 16

- ADAPTER LTH density: $d_{\mathrm{LTH}} \in [0.97, 0.98, 0.99]$ (based on num. of rounds $\in [100, 200, 400]$ respectively) We prune each round for all datasets besides FLAIR, and prune once every 25 rounds for FLAIR with density 0.98.

- SPARSEADAPTER density $d$: 1/4

- **FLASC** density: $p_{\mathrm{down}} = d_{\mathrm{up}} \in \mathbf{1/4}$. For 20NewsGroups and Reddit, we set $d_{\mathrm{down}} = 1$.

- Label heterogeneity $\alpha$: CIFAR10: 0.1, 20NewsGroups: 0.01

Figure 4 (Reddit):

- ADAPTER LTH density: $p_{\mathrm{LTH}} = 0.99$

- SPARSEADAPTER density: $p = 1/4$

- **FLASC** density: $p_{\mathrm{down}} = 1/4, p_{\mathrm{up}} \in [1/16, 1/4]$

Figure 5 (CIFAR10):

- LoRA rank: $r = 16$

- Density (all methods): $d \in [1, 1/4, 1/16, 1/64, 1/256]$

Figure 7 (20NewsGroups):

- SLoRA Stage 1 LR: 1e-3

- Stage 1 rounds: 50 (25% of 200 total rounds)

Table 3 (20NewsGroups): Details in text of Sec. 4.4.

Figure 8 (FLAIR):

- Noise multiplier: $\sigma \in [0, 0.34]$

- Clipping norm: $C = 5 * 10^{-3}$

Figure 8,9 (Reddit):

- Client learning rate: $\eta_{\text{client}} = 5 * 10^{-4}$

- Noise multiplier: $\sigma \in [0, 0.013, 0.072, 0.58]$

- Clipping norm: $C = 10^{-4}$

### B.5   Tabular Results

For completeness, we provide the exact numbers from all the figures besides those where such numbers are already provided (Figures 1, 4, 8 (right) and Tables 1, 3).

|  | CIFAR10 | 20NewsGroups | Reddit ($\downarrow$) | FLAIR |
|---|---|---|---|---|
| Upload budget | 100MB | 230MB | 230MB | 2300MB |
| Performance at 20% of budget | | | | |
| LoRA | 89.6 | 17.2 | 52.8 | 64.9 |
| Adapter LTH | 85.4 | 46.5 | 50.2 | 60.8 |
| SparseAdapter | 83.4 | 51.5 | 52.3 | 55.8 |
| **FLASC** | **97.2** | **65.6** | **49.4** | **73.9** |
| Performance at 100% of budget | | | | |
| LoRA | 97.7 | 66.5 | 48.8 | 76.3 |
| Adapter LTH | 97.5 | 73.3 | 48.0 | 74.9 |
| SparseAdapter | 85.5 | 79.6 | 49.0 | 60.2 |
| **FLASC** | **98.0** | **80.8** | **47.6** | **77.9** |

**Table 10:** Tabular results for accuracy (perplexity for Reddit) from Figure 3.

| Density | 1/4 | 1/16 | 1/64 | 1/256 |
|---|---|---|---|---|
| IID ($\alpha = 100$) | | | | |
| LoRA (density=1) | | 99.0 | | |
| Federated Select | 85.0 | 70.7 | 62.8 | 61.3 |
| SparseAdapter | 86.5 | 81.1 | 78.3 | 76.9 |
| **FLASC** | **98.1** | **98.1** | **97.9** | **88.4** |
| NIID ($\alpha = 0.01$) | | | | |
| LoRA (density=1) | | 98.5 | | |
| Federated Select | 83.0 | 67.8 | 58.2 | 56.4 |
| SparseAdapter | 85.7 | 80.1 | 77.2 | 75.1 |
| **FLASC** | **97.6** | **97.0** | **97.4** | **86.1** |

**Table 11:** Tabular accuracy results for CIFAR10 from Figure 5.

| Rank | Density | | | |
|------|-----|-----|-----|-----|
| ↗ | 1/8 | 1/4 | 1/2 | 1 |
| 1 | 30.5 | **32.5** | **39.4** | 39.3 |
| 2 | **26.4** | **50.1** | 51.8 | 50.1 |
| 4 | 32.5 | **57.1** | 62.6 | 61.7 |
| 8 | 43.9 | **63.3** | 62.8 | 63.0 |
| 16 | 45.6 | **63.5** | 63.0 | 61.7 |
| 32 | 50.8 | **62.8** | 64.5 | 63.9 |
| 64 | 57.9 | **63.9** | 62.9 | 63.4 |

**Table 12:** Tabular Subreddits accuracy results from Figure 6 (left). Cells along a common diagonal (NE-SW) have the same communication cost. Bolded cells are searched by **FLASC-S**.

| Upload budget | 0.5GB | 1GB | 2GB | 4GB | 8GB | 16GB |
|---------------|-------|------|------|------|----------|------|
| LoRA | 39.3 | 39.3 | 50.1 | 61.7 | 63.0 | 63.0 |
| **FLASC** | 39.4 | 39.4 | 51.8 | 57.1 | 63.3 | 63.5 |
| Adapter LTH | 35.8 | 39.3 | 55.8 | 57.5 | **64.0** | 64.0 |
| **FLASC-S** | **50.1** | **57.1** | **63.3** | **63.5** | 63.5 | **64.0** |

**Table 13:** Tabular Subreddits accuracy results from Figure 6 (right).

| | IID, $\alpha = 100$ | | | NIID, $\alpha = 0.1$ | | | Upload cost (GB) | | |
|------|------|------|------|------|------|------|------|------|------|
| Rank | 1 | 4 | 16 | 1 | 4 | 16 | 1 | 4 | 16 |
| LoRA | 77.0 | 79.9 | 80.5 | 68.4 | 75.7 | 77.8 | 0.3 | 1.2 | 4.7 |
| SLoRA | 74.9 | 75.6 | 77.8 | 60.4 | 62.4 | 65.1 | 4.5 | 5.1 | 7.8 |
| FLoRA | 75.0 | 77.4 | 80.2 | 61.0 | 71.1 | 74.7 | 42.7 | 43.4 | 46.0 |
| **FLASC** | 76.8 | 80.0 | 79.9 | 61.7 | 73.6 | 77.0 | 0.1 | 0.3 | 1.2 |

**Table 14:** Tabular 20NewsGroups accuracy results from Figure 7.

| Upload budget | 0.5GB | 1GB | 2GB | 4GB | 8GB | 16GB |
|---------------|-------|------|------|------|----------|------|
| LoRA | 39.3 | 39.3 | 50.1 | 61.7 | 63.0 | 63.0 |
| **FLASC** | 39.4 | 39.4 | 51.8 | 57.1 | 63.3 | 63.5 |
| Adapter LTH | 35.8 | 39.3 | 55.8 | 57.5 | **64.0** | 64.0 |
| **FLASC-S** | **50.1** | **57.1** | **63.3** | **63.5** | 63.5 | **64.0** |

**Table 15:** Tabular Subreddits accuracy results from Figure 6 (right).

| Non-private ($\epsilon = \infty$) | | | | |
|---|---|---|---|---|
| Upload budget (MB) | 100 | 200 | 300 | 400 |
| LoRA | 49.8 | 49.3 | 48.1 | 47.9 |
| FFA-LoRA | 52.2 | 50.9 | 50.1 | 49.9 |
| **FLASC** | **48.1** | **46.8** | **46.8** | **46.4** |
| Private ($\epsilon = 1$) | | | | |
| Upload budget (MB) | 50 | 100 | 150 | 200 |
| LoRA | 56.2 | 54.9 | 54.0 | 53.6 |
| FFA-LoRA | 59.0 | 57.8 | 56.4 | 55.8 |
| **FLASC** | **53.4** | **52.7** | **52.5** | 52.9 |

**Table 16:** Tabular Reddit perplexity ($\downarrow$) results from Figure 8.

| Rank | 1 | 4 | 16 | 64 |
|---|---|---|---|---|
| Non-private ($\epsilon = \infty$) | | | | |
| LoRA | 48.4 | 47.9 | 47.8 | **47.2** |
| FFA-LoRA | 50.5 | 50.0 | 49.9 | 49.4 |
| **FLASC** | 49.0 | 48.7 | 48.0 | 47.6 |
| Private ($\epsilon = 1$) | | | | |
| LoRA | 53.4 | 52.6 | **52.2** | 52.7 |
| FFA-LoRA | 53.8 | 53.5 | 54.5 | 56.1 |
| **FLASC** | 54.0 | 53.0 | 52.5 | 52.9 |

**Table 17:** Tabular Reddit perplexity ($\downarrow$) results from Figure 9. Note that for the same rank, FFA-LoRA uses 75% of the communication of LoRA, while **FLASC** uses 25% ($3\times$ less than FFA-LoRA).

