# OpenReview forum: "FLASC: Federated LoRA with Sparse Communication"
_TMLR — Rejected by TMLR_

### Review · Reviewer_eDpA · 2025-06-07

**Summary Of Contributions:**

The paper proposes the FLASC framework, which introduces a strategy of *dense local fine-tuning* combined with *Top-K unstructured sparse communication during uplink*. The method claims to reduce LoRA upload cost by up to **10×** or improve accuracy by 20% under the same communication budget, all without loss of accuracy. Extensive experiments are conducted across multiple aggregated datasets, with evaluations under  data/system heterogeneity and differential privacy settings.

**Audience:**

Yes

**Claims And Evidence:**

Yes

**Requested Changes:**

1. Figure 6 is hard to understand, especially the right subfigure. It lacks essential legends or axis explanations, and there is **no sufficient clarification in the text**.
   Also, FLASC appears to be a heuristic-based strategy, but it is not clearly explained. Please clarify its realization clearly and whether a **new dataset would require a new search process** for optimal sparsity.

2. In Section 4.5 on Privacy, the introduced FFA-LoRA underperforms compared to vanilla LoRA. This contradicts the original paper’s conclusion. Please provide a clear interpretation or justification.

3. Several notations are undefined**causing confusion:
   - `r_base` in Section 4.4
   - `ε (epsilon)` in Section 4.5
   - The meaning of **"budget tier"** is also unclear in Section 4.4.

    It is recommended to review the entire manuscript for missing symbol definitions and restructure Section 4.4 to improve readability and comprehension.

**Strengths And Weaknesses:**

###  Strengths

- **Simple yet effective**: The method only modifies the upload phase and is easy to integrate into existing FL systems.
- **Robust performance**: Demonstrates consistent improvement across multiple datasets.

###  Weaknesses

- While the motivation is valid, the proposed idea seems to overlap significantly with* the approach in *"Achieving Personalized Federated Learning with Sparse Local Models"*. This raises concerns about the  contribution.
   I strongly encourage*the authors to clearly explain the distinction between FLASC and the prior work, ideally with an explicit discussion in the main paper.

---

> ### Author Response · Authors · 2025-07-08
>
> We thank the reviewer for recognizing that FLASC is a simple yet effective method, which is a central aspect of our paper.
>
> We have split our response into two parts due to the character limit.
> **1. Distinction between FLASC and FedSpa [1].**
>
> There are a number of key differences between FLASC and FedSpa; we clarify these below and have revised the paper to include this discussion. For completeness, we also include a comparison to FedSpa in Table 1 alongside other backbone pruning methods. Given the poor performance of FedSpa under our constrained setup, we include an additional ablation in **Table 4** of the appendix which allocates more training rounds.
>
> FedSpa is a “sparse backbone” method introduced in [1]. FLASC and FedSpa are indeed similar in that both methods communicate sparse updates and these updates may have different sparsity patterns across clients. However, there are a number of key differences in the contribution of FLASC versus FedSpa:
> 1. **Model parameterization.** FedSpa falls under the category of “sparse backbone” methods, i.e., the backbone parameters are the ones being finetuned and sparsified. As we show in Table 1, methods that follow this approach perform worse than simply using LoRA due to the importance of the pretrained weights in our finetuning setting.
> 2. **Application of sparsity.** In FedSpa, clients locally finetune a sparse backbone with a fixed mask. After each local finetuning round, a “mask searching” method adjusts the mask. In contrast, in FLASC, clients locally finetune a dense LoRA adapter and only apply Top-K sparsity to communication. Compared to FedSpa, FLASC does not need to statefully store the client masks and is better suited for settings with a large number of clients.
> 3. **Final model.** FedSpa produces a personalized sparse backbone for each client, while FLASC produces a global LoRA adapter. While personalization may improve accuracy, the single global model produced by FLASC is applicable to more problems (e.g. evaluating on unseen clients or a global dataset).
>
> Overall, our work crucially shows that FLASC is an overlooked baseline for communication-efficient FL. Like many existing sparse FL methods, FedSpa applies sparsity during local finetuning, but FLASC shows that this complexity is not necessary. To address the main goal of reducing communication, we find that it is adequate to use a simple combination of (1) parameter-efficient finetuning and (2) sparse communication.
>
> [1] Huang, Tiansheng, et al. "Achieving personalized federated learning with sparse local models." arXiv preprint arXiv:2201.11380 (2022).
>
> **2. Figure 6 is hard to understand.**
>
> Figure 6 considers two key hyperparameters of FLASC: (1) LoRA rank and (2) upload density. We agree that the dependence on the density hyperparameter can be made more clear and have revised the figure to better show this. We have also updated the text in Section 4.2 to more clearly explain the figures.
>
> **3. FLASC appears to be a heuristic-based strategy, but it is not clearly explained.**
>
> FLASC, like all other LoRA / sparse training methods, requires the user to tune the rank and/or sparsity. Generally, a new dataset would require a new search process for optimal hyperparameters, and FLASC is no exception. Even for methods which “automatically” adjust these hyperparameters, there are other new hyperparameters to tune, e.g., upper/lower bounds on rank/sparsity and frequency/size of the adjustments.
>
> In Section 4.2, precisely due to these HP tuning dependencies, we show that it is important to account for hyperparameter tuning when comparing sparsity-based approaches to regular LoRA. This is because the additional cost of searching over sparsity values can negate the benefits of applying sparsity in the first place. Therefore, we present FLASC-S, a method that efficiently searches over FLASC hyperparameters by first tuning sparsity on a small rank, and then tuning the rank with this fixed sparsity.
>
> Overall, we found that a density of 0.25 is robust across the four datasets we tested. Applying this to new datasets would provide the same communication savings (75% per-round reduction compared to LoRA), but may leave a small amount of communication savings on the table if the optimal density is lower.

---

> > ### Author Response · Authors · 2025-07-08
> >
> > **4. FFA-LoRA underperforms compared to vanilla LoRA.**
> >
> > Thanks for pointing this out. The main difference between the experiments in the FFA-LoRA paper versus ours is that FFA-LoRA experiments use FedAvg, while FLASC experiments use FedAdam. FedAdam is a state-of-the-art FL optimizer which is commonly used in practice and generally outperforms FedAvg [2].
> >
> > Since the FFA-LoRA code is not public, we attempted to reproduce their experimental setup on the QNLI dataset and RoBERTa-Large model. Following the setup in Table 2 (i.i.d) of the FFA-LoRA paper, we split QNLI into 3 i.i.d. partitions and train for 1000 rounds for 10 local steps at each round. **Our results are presented in **Table 5** of the appendix in the updated draft.**
> >
> > With FedAvg, FFA-LoRA slightly improves over LoRA when using a large batch size and significantly improves when using a small batch size. We hypothesize that LoRA can fail to train in a small-batch setting due to noise from stochastic gradients compounding with the inexact aggregation of LoRA. However, FedAdam appears to mitigate these issues. Using FedAdam with a small batch size (16), both LoRA and FFA-LoRA outperform their FedAvg counterparts and LoRA performs better than FFA-LoRA.
> >
> > [2] Reddi, Sashank J., et al. "Adaptive Federated Optimization." International Conference on Learning Representations.
> >
> > **5. Several notations are undefined.**
> >
> > Thanks for pointing these out. We have clarified these notations (and others) in the text, and have reorganized Section 4.4 for improved readability. We have highlighted our larger updates in red.

---

> > > ### Comment · Reviewer_eDpA · 2025-07-25
> > > **Replying to Official Comment by Authors**
> > >
> > > I appreciate the authors' efforts in providing a detailed response. It has satisfactorily addressed most of my concerns.

---

### Review · Reviewer_Gm4b · 2025-07-06

**Summary Of Contributions:**

The authors introduce a Federated finetuning method FLASC which combined LoRA update at the client level with sparse communication with the server. While LoRA update ensure efficient client level updates, a sparse mask reduces the communication overhead to the server compared to other baselines. The proposed method can thus be communication efficient while increasing expressivity at the client level. This translates to reduced computation cost for achieving similar accuracies compared to baselines.

**Audience:**

Yes

**Claims And Evidence:**

Yes

**Requested Changes:**

The paper is well written and the authors provide experimental results to verify their claims.

**Strengths And Weaknesses:**

Strengths
1. The authors propose an efficient Federated finetuning baseline which leverages LoRA adapters at the client level and a sparse update communicated to the server to reduce communication costs, thus improving communication vs accuracy tradeoff, without compromising on expressivity of the client compared to other baselines.
2. Experimental results show that the proposed method serves as a strong baseline, even across heterogenous datasets.
3. The authors also provide a tuning strategy to find the appropriate rank versus density ratio which is more sophisiticated than a grid search and makes the method more user friendly.

Weaknesses
1. While the authors leverage simple pruning algorithms to find the sparse mask, the structure of the mask can further help reduced communication costs, for example a structured mask. It would be additionally useful to have a discussion on the structure of the mask and its communication cost.

---

> ### Author Response · Authors · 2025-07-08
>
> We thank the reviewer for recognizing key strengths of FLASC, namely its simplicity, robustness across many settings, and user-friendly tuning strategy.
>
> **It would be additionally useful to have a discussion on the structure of the mask and its communication cost.**
>
> We agree and have revised the paper to include the discussion below on how certain factors affect the communication cost of the mask.
>
> **Mask structure.** As mentioned in Section 3, we use **unstructured** sparsity which operates on individual model weights. Since the mask is determined with a Top-K function, the mask is a binary vector with the same size as the LoRA parameters. Grouping parameters into larger structures (e.g. columns, rows, or blocks) can indeed reduce the communication cost of the mask itself by only requiring one bit to indicate whether a group of parameters is masked. However, an unstructured approach allows for more expressive updates, an advantage we discuss in Section 2 (“Sparsity and pruning”).
>
> **Sparsity format.** For the purposes of communication, the mask is stored as a binary vector with $p$ entries where $p$ is the number of trainable parameters in the model. The cost of communicating this mask is (1/32)*(4 bits per parameter)*$p$, which is independent of the actual amount of sparsity in the mask. We use this format for simplicity and because FLASC (like other model sparsity methods) requires relatively large density values. If communication is sufficiently sparse, it would be more beneficial to use a sparse matrix format which scales with sparsity. Such a matrix format (e.g. CSR) stores only the non-zero indices but each index (e.g. uint8) requires multiple bits.
>
> Please let us know if this addresses your question. We are happy to provide further clarification.

---

> > ### Author Response · Authors · 2025-07-09
> >
> > We would like to correct a typo in the above response: the cost of communicating the mask is simply $p$ bits; $(4/32)p = (1/8)p$ would be the size in bytes.

---

> > > ### Comment · Reviewer_Gm4b · 2025-08-04
> > > **Response to Authors**
> > >
> > > I thank the authors for providing clarifications, my concerns have been largely addressed.

---

### Review · Reviewer_Xudz · 2025-07-24

**Summary Of Contributions:**

This paper proposes FLASC, a method to improve communication efficiency in FL by applying Top-K sparsification to the LoRA update during the upload stage. The core idea is to keep local training dense while sparsifying only the updates transmitted to the server. The paper also introduces FLASC-S, a simple yet efficient hyperparameter tuning strategy to jointly select sparsity and rank. The authors evaluate their method on several datasets and tasks, demonstrating that FLASC achieves comparable or better performance than standard LoRA while significantly reducing communication overhead.

**Audience:**

No

**Broader Impact Concerns:**

The paper does not raise any immediate ethical or societal concerns.

**Claims And Evidence:**

No

**Requested Changes:**

1. The current baselines do not allow for a fair evaluation of FLASC's communication efficiency. Either compare to existing PEFT methods specifically designed for FL, or clearly state the lack of such baselines and the limitations of current comparisons.

**Strengths And Weaknesses:**

**Strengths**

1. Even though LoRA is already more communication-efficient than full model fine-tuning, further improving the communication efficiency of LoRA remains a valuable and practically relevant goal, especially in real-world federated learning scenarios with strict bandwidth limitations.

**Weakness**

1. The core technique—using Top-K sparsification to reduce communication in FL—is a standard and widely-used idea in the distributed learning literature (e.g., gradient sparsification, Deep Gradient Compression, TernGrad). Applying Top-K sparsification to LoRA updates is a natural extension, and its conceptual contribution is minimal. Using Top-K to reduce communication can be considered common knowledge in distributed learning and should not be regarded as a methodological innovation.

2. Many of the baselines used in the paper (FedSpa, FedSparsify, SPDST, Federated Select) are based on full model fine-tuning, which is inherently more communication-intensive than PEFT methods like LoRA. It is not surprising that FLASC outperforms them in communication efficiency. Other baselines like Adapter LTH and SparseAdapter, while based on LoRA, are designed for centralized training settings and do not consider communication costs at all. Comparing FLASC to these methods on communication metrics is not meaningful or fair, since they were never designed with communication costs in mind.

---

> ### Author Response · Authors · 2025-08-05
>
> **We appreciate the reviewer’s feedback and have made several revisions in Sections 2 and 3. Please let us know if this adequately addresses your concerns, especially the concerns of evaluation.** While we already compare FLASC to a set of representative methods (see W2 below), we are happy to discuss any additional suggestions for evaluation.
>
> **W1. Applying Top-K sparsification to LoRA updates is a natural extension, and its conceptual contribution is minimal.**
>
> We agree that FLASC is a natural extension, and that Top-K sparsification itself is common knowledge. In our view, this makes it even more surprising that FLASC has been overlooked as a simple and effective baseline for FL. FLASC fills in two important missing pieces in the current literature: First, sparse communication has been widely discussed in FL, but its application to LoRA has not been well-studied. Second, while many extensions to LoRA have been proposed in centralized and FL settings, we show that simple TopK sparsity is more communication-efficient than these complex methods. **We have revised Section 2 and Section 3 to emphasize that FLASC’s simplicity is an important contribution.**
>
> **In addition to addressing these limitations, our work points out that “sparse LoRA” methods introduce additional costs for hyperparameter tuning.** Therefore, we propose FLASC-S, a novel hyperparameter tuning method that allows us to retain the benefits of FLASC even when considering the costs of hyperparameter tuning.
>
> **W2. Comparing FLASC to these methods on communication metrics is not meaningful or fair, since they were never designed with communication costs in mind.**
>
> **We agree that it is not surprising that FLASC outperforms the full model baselines.** The purpose of these experiments is to show that LoRA *alone* performs better than such baselines and to establish LoRA as a strong starting point (something that has been overlooked by a subset of prior work in communication-efficient FL).
>
> **AdapterLTH and SparseAdapter are reasonable baselines.** While the reviewer is correct that AdapterLTH and SparseAdapter were not designed with communication costs in mind, these two LoRA methods (1) leverage the same key ideas as full model methods FedSparsify (iterative magnitude pruning) and SPDST (pruning at initialization) and (2) are more similar to FLASC than the FL-specific methods we consider. To the best of our knowledge, no FL-specific work has proposed applying weight-level sparsity to LoRA as we do in FLASC. **We have revised Section 3 (“LoRA is a strong baseline”) to clarify these points.**
>
> **We do compare to FL-specific LoRA baselines.** We not only compare FLASC to full model finetuning and centralized LoRA approaches (Section 3 and 4.1), but also to SLoRA (Section 4.3), HetLoRA (Section 4.4), and FFA-LoRA (Section 4.5). **To make this more clear, we have added a discussion of these methods in Section 3 (“FL LoRA Baselines”).**

---

### Review · Reviewer_t3rX · 2025-07-24

**Summary Of Contributions:**

This paper introduces LASC: Federated LoRA with Sparse Communication, a  approach addressing critical communication efficiency and hyperparameter tuning challenges for LoRA-based fine-tuning in cross-device Federated Learning (FL) settings. FLASC  enables clients to perform dense local fine-tuning and only sparsifying updates during upload. Complementing this, the author propose FLASC-S (FLASC-Search), a hyperparameter search strategy that intelligently search the rank and sparsity configuration space,  reducing overall tuning overhead compared to traditional grid search.

**Audience:**

No

**Broader Impact Concerns:**

The paper does not raise any immediate ethical or societal concerns.

**Claims And Evidence:**

No

**Requested Changes:**

Refer to the ​Weaknesses section, particularly the experimental presentation and related work discussions

**Strengths And Weaknesses:**

Strength:

- **Novel and Effective Core Methodology:** FLASC performs dense local fine-tuning while  sparsifying updates during communication, is a simple and intuitive design.
- **Communication Efficiency Improvement:** The experiments demonstrate FLASC's advantage in reducing communication overhead. It achieves up to 10x less communication for comparable accuracy or up to 20% higher accuracy under the same communication budget.
- **Hyperparameter Tuning Strategy:** The proposed FLASC-Search employs a clever sequential tuning strategy (density first, then rank) . This pragmatic approach significantly enhances FLASC's practical appeal.


Weakness:


- **Simplicity of TopK Mask and Potential Limitations:** While FLASC demonstrates strong  performance with a straightforward l1-norm based TopK operation for mask selection, Further exploration or theoretical justification for its robustness would strengthen the methodological underpinnings. Additionally, it remains underexplored whether more sophisticated sparsification strategies (e.g., those considering gradient direction, historical information, or other importance metrics) could yield further performance gains or prove superior in specific, challenging scenarios.
- **Simplified Analysis of Compute Costs:** The paper states that FLASC maintains the same computational costs as LoRA, attributing this to the dense local training and asserting that LoRA adapter's compute and memory overhead is negligible compared to the backbone. A more detailed quantitative analysis of client-side computational load, particularly in the context of varying model sizes and across different degrees of system heterogeneity, would provide a better understanding
- **Poor experiments presentation**: Although the extensive experiments are conducted, most of the results is figure, lakc the accurate quanlitative results. This hinder the experiment representation and lack of methodological rigor.
- **Limited discussion of Diverse Sparsity Patterns:** The author should discuss whether FLASC's framework could be extended to accommodate other hybrid sparsity patterns, or under what specific conditions structured sparsity might become advantageous within this federated context. A brief discussion on this could enrich the scope.
- **Lack latest related work**: The author shoud discuss more latest work in related work:
    - GSQ-Tuning: Group-Shared Exponents Integer in Fully Quantized Training for LLMs On-Device Fine-tuning, ACL 2025.

---

> ### Comment · Reviewer_t3rX · 2025-08-05
> **Does authors have any updates for my concerns?**
>
> Dear authors,
>
> DO you have any updates or rebuttals on my concerns?

---

> > ### Author Response · Authors · 2025-08-05
> >
> > Yes, thanks for reaching out. We posted a response and updated the paper a few minutes ago.

---

> ### Author Response · Authors · 2025-08-05
>
> **We appreciate the reviewer’s feedback and have added two new appendix sections (Appendix A.2, B.5) in response. Please let us know if this adequately addresses your concerns.** Namely, we provide all the exact numbers from the figures as tables (W1), cite more recent works (W5), and provide additional discussion and experiments about the TopK mask (W1), compute cost (W2), and structured sparsity (W4).
>
> **W1. Simplicity of TopK Mask and Potential Limitations.**
>
> We appreciate the reviewer for noting that the L1 norm of the delta (local weight update) is a simple and effective criteria for mask selection. Theoretically, our goal is to sparsify the delta while remaining close to the updated model, which makes the delta L1 (Algorithm 1, L17) the most natural choice. More complex importance metrics are either:
> - **impractical:** information from previous rounds is not available in stateless cross-device settings,
> - **already encoded into the delta:** gradients from local steps are incorporated during finetuning,
> - **or perform worse:** a weight may be updated significantly but end up in a region with a small gradient.
>
> To validate this intuition, we ran an additional experiment which shows that delta L1 performs better than two more complex baselines: gradient and gradient-delta product. **We have included this discussion and experiment in Appendix A.2 (“Sparsity criteria”).**
>
> **W2. Simplified Analysis of Compute Costs.**
>
> **We provide an exact analysis on the FLOPs in Appendix A.2 (“Compute benefit of sparsity”).** If we consider larger models, the relative cost of LoRA becomes even smaller, as the number of backbone parameters scales quadratically with width while the number of LoRA parameters scales linearly. While analyzing compute cost under systems heterogeneity is interesting, this is an orthogonal question; the speed of the client does not affect the relative cost of the backbone versus LoRA. However, as shown by this experiment, simply adjusting the rank (or applying sparse LoRA) does not do much to change the compute cost; therefore, when we discuss systems heterogeneity, we focus on different communication budgets across clients.
>
> **W3. Poor experiments presentation.**
>
> We provide specific numbers in several figures (Figures 1, 4, 8) and tables (Tables 1 and 3). The reason we provide several results in the form of line charts is because they show a tradeoff between communication and accuracy. We do not believe this harms the methodological rigor of our experiments, but rather helps the reader more clearly understand the results. **However, in light of the reviewer’s suggestion, in Appendix B.5, we provide the specific numbers for all remaining figures (3,5,6,7,9) in table form.**
>
> **W4. Limited discussion of Diverse Sparsity Patterns**
>
> FLASC can indeed be extended to accommodate structured sparsity patterns. However, as we discuss in Section 2, an unstructured approach allows for the most expressive updates and best utility. To support this, we ran an experiment which shows that (1) structured sparsity saves only a small amount of total communication and (2) significantly harms model training. Structured sparsity can be effective if some features are naturally sparse for all data points at a given client, but this is unlikely in practice. **We have included this discussion and experiment in Appendix A.2 (“Sparsity pattern”).**
>
> **W5. Lack latest related work**
>
> We are happy to cite the GSQ-Tuning work. **Since this work applies quantization to LoRA in centralized settings, we have included it in Section 2 (“Efficient LoRA”).** Beyond this paper, we've also cited several other recent papers, including some that came out after our paper was submitted to TMLR (see below).
>
> Zhou, Sifan, et al. "GSQ-Tuning: Group-Shared Exponents Integer in Fully Quantized Training for LLMs On-Device Fine-tuning." arXiv preprint arXiv:2502.12913 (2025).
>
> Singhal, Raghav, Kaustubh Ponkshe, and Praneeth Vepakomma. "FedEx-LoRA: Exact Aggregation for Federated and Efficient Fine-Tuning of Large Language Models." Proceedings of the 63rd Annual Meeting of the Association for Computational Linguistics (Volume 1: Long Papers). 2025.
>
> Bossy, Thierry, et al. "Mitigating Unintended Memorization with LoRA in Federated Learning for LLMs." ICML 2025 Workshop on Collaborative and Federated Agentic Workflows.
>
> Zhang, Juzheng, et al. "Lori: Reducing cross-task interference in multi-task low-rank adaptation." arXiv preprint arXiv:2504.07448 (2025).
>
> Chen, Zhijie, Yuxing Liu, and Arindam Banerjee. "Truncate without Fear: Module Aggregation and Redistribution in Federated Low-Rank Adaptation." ICLR 2025 Workshop on Modularity for Collaborative, Decentralized, and Continual Deep Learning.

---

> > ### Comment · Reviewer_t3rX · 2025-09-05
> > **My concerns have been solved**
> >
> > Thanks to the author's detailed rebuttal and corresponding revision, my concern has been solved. The revisions need to be included in the final version.

---

### Author Response · Authors · 2025-08-05

We would like to thank all the reviewers for their feedback and suggestions. **We have responded to each reviewer individually and would like to use this global response to justify how our work has (1) reasonable claims supported by evidence and (2) relevance to TMLR’s audience.**

**Claims and supporting evidence**

The major concerns reviewers raised were that (1) the method is simple (Xudz, t3rX), (2) some baselines are weak (Xudz), and (3) some details require more discussion and analysis (eDpA, Gm4b, Xudz, t3rX).

While these concerns are legitimate, they do not affect the central claims of the paper, nor do they indicate that there is insufficient evidence for these claims. After these revisions, the three main claims of our paper remain unchanged: LoRA itself is a strong baseline for communication-efficient FL (Section 3), FLASC improves the communication efficiency of LoRA (Section 4.1), and FLASC is robust to several other concerns in FL (Sections 4.2, 4.3, 4.4, and 4.5). As the reviewers noted, we conducted extensive experiments (eDpA, t3rX) over several datasets (Gm4b, Xudz) to validate these claims.

The paper acknowledges and addresses the first two concerns:

(1) Section 1: “Despite its simplicity, FLASC has been overlooked as a baseline by prior work in FL.” (We believe that the simplicity of the method is in fact a key advantage and makes it more likely to be used in practice.)

(2) Section 3: “Since sparse backbone methods are unable to compete with LoRA, we only consider LoRA and sparse LoRA methods as baselines throughout this work.” (We agree that sparse backbone methods are weak baselines; the purpose of evaluating them is to justify combining sparsity with LoRA, as it is natural for readers to wonder which of the two methods is more effective on its own. **To make this more clear, we have revised this section.**)

(3) For more specific concerns, we have responded to individual reviewers and added new discussions and experiments in the paper.

**Relevance and interest to TMLR’s audience**

We believe a simple yet overlooked baseline for communication-efficient FL would be of interest to TMLR’s audience. Many works published at TMLR focus precisely on communication-efficient FL. We provide a few examples below.

Zaccone, Riccardo, et al. "Communication-Efficient Heterogeneous Federated Learning with Generalized Heavy-Ball Momentum." Transactions on Machine Learning Research (2023).

Tobaben, Marlon, et al. "NeurIPS 2023 Competition: Privacy Preserving Federated Learning Document VQA." Transactions on Machine Learning Research (2025).

Bergou, El Houcine, et al. "Personalized Federated Learning with Communication Compression." Transactions on Machine Learning Research (2023).

Babakniya, Sara, et al. "Revisiting sparsity hunting in federated learning: Why does sparsity consensus matter?." Transactions on Machine Learning Research (2023).

Niu, Yue, et al. "Overcoming resource constraints in federated learning: Large models can be trained with only weak clients." Transactions on Machine Learning Research (2023).

---

### Decision · Action_Editor_QpM2 · 2025-09-04

**Recommendation:** Reject

**Additional Comments:**

The paper at this stage lacks accurate, convincing and clear evidence to support the major claim of the submission. More experimental results are needed if the authors plan to resubmit to TMLR with major revision.

**Audience:**

Yes

**Audience Explanation:**

All reviewers agree that the problem communication-efficient FL training (especially Lora) is important and practical.

**Claims And Evidence:**

No

**Claims Explanation:**

In this submission, the authors introduced a Top-K sparsification algorithm that applied to LoRA updates under the federated learning setting. All reviewers agree that the problem is important and practical. However,  reviewers still have concerns (after rebuttal) about the evaluations at this stage, which cannot support the claim of "Top-K on LoRA reduces bytes".

**Resubmission Of Major Revision:**

The authors may consider submitting a major revision at a later time.